# MITIGATING THE ECHO CHAMBER EFFECT IN KV CACHE COMPRESSION VIA COVERAGE OPTIMIZATION

## ABSTRACT

Large language models (LLMs) have achieved strong performance on complex tasks ranging from multi-document reasoning to long-dependency question answering. To enable efficient inference, these models rely on key-value (KV) caching, which stores and reuses KV pairs to avoid redundant computation. As the sequence length grows, the KV cache increases linearly, creating a severe GPU memory bottleneck. This issue is commonly addressed by compressing the KV cache using a top-k selection based on attention scores. However, this strategy induces a homogeneity bias, the tendency to repeatedly select similar tokens, which creates an Echo Chamber Effect where the compressed KV cache is dominated by redundant information. This results in low effective coverage, causing crucial information to be lost and leading to verbose and logically broken answers under constrained token budgets. To address this, we propose ApertureKV, a KV cache compression method that employs coverage optimizing strategies to mitigate the Echo Chamber Effect. ApertureKV addresses two distinct sources of redundancy through two core components: Query Diversification (QD), which adjusts queries to encourage the retention of a more diverse set of tokens, and Redundancy-Aware Budget Allocation (RABA), which allocates more budget to heads that capture distinct information. By achieving highly effective coverage, ApertureKV enables robust KV cache compression under tight memory constraints, yielding more accurate responses. Evaluations on long-context benchmarks such as LongBench and LooGLE, including Needle-in-a-Haystack tasks, show that ApertureKV consistently outperforms state-of-the-art methods under tight budgets. In particular, on one LongBench sub-task with Mistral-7B-Instruct, ApertureKV retains 92.6% of FullKV performance while using only 0.2% of the KV cache budget.

## 1 INTRODUCTION

Large language models (LLMs) (Anthropic, 2024; Team et al., 2024; Grattafiori et al., 2024; Guo et al., 2024; DeepSeek-AI, 2025) achieve strong performance on diverse NLP tasks, and their efficiency in long-context inference critically depends on key-value (KV) caching. KV caching stores intermediate attention states to avoid repeated computation, enabling high throughput for tasks such as multi-document reasoning (Bai et al., 2024), long-dependency question answering (Li et al., 2024a; Zhu et al., 2024; Wang et al., 2025). In practice, LLM inference consists of two stages: during the prefill stage, the model constructs the KV cache for the input prompt, and during the decode stage, it generates tokens by updating and reusing KV cache. Through this mechanism, KV caching has become a key technique for efficient long-context inference and a standard component of modern LLM serving pipelines.

However, the size of the KV cache increases linearly with sequence length, since each newly generated token produces a key and value vector that is appended to the KV cache at every attention layer. This linear growth results in a significant GPU memory bottleneck and, in constrained environments, can rapidly exhaust the available capacity. Such overhead not only restricts the effective context length that can be maintained but also reduces the degree of parallelism achievable in serving systems. The problem is further intensified in small-scale platforms such as mobile or edge devices, where memory resources are severely limited. These limitations highlight the necessity of methods that reduce the KV cache size while preserving model accuracy.

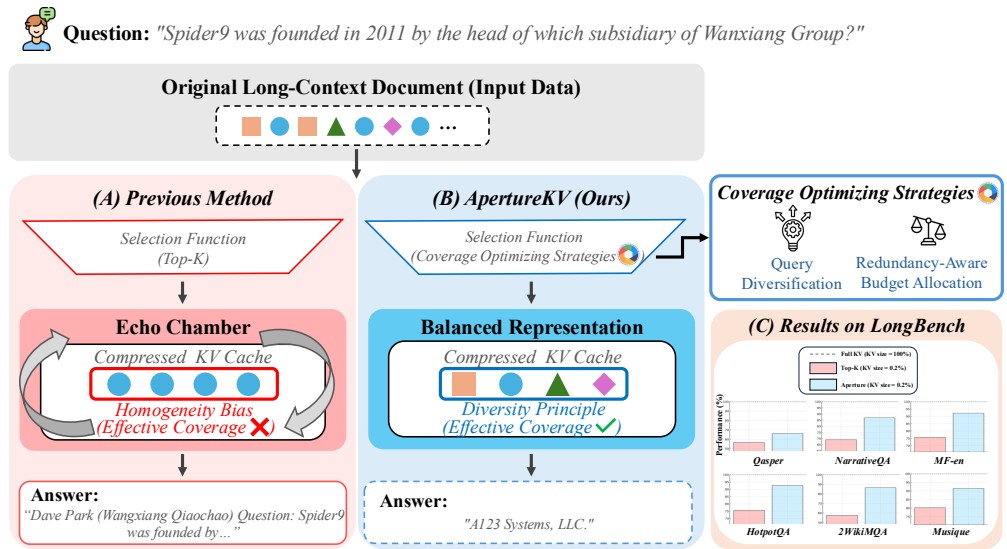

Figure 1: **The Echo Chamber Effect in KV Cache Compression.** (A) Previous Method suffers from homogeneity bias, forming an Echo Chamber that yields redundant KV caches, low effective coverage, and verbose answers. (B) ApertureKV applies coverage-optimizing strategies guided by the diversity principle, producing balanced KV caches, high effective coverage, and accurate answers. (C) On six LongBench sub-tasks under a constrained budget (0.2% of the KV cache), ApertureKV consistently outperforms previous method and significantly narrows the gap to Full KV.

To address this issue, existing methods (Zhang et al., 2023; Feng et al., 2024; Li et al., 2024b; Cai et al., 2024) typically apply KV cache compression by using a selection function to retain a subset of tokens based on top-k attention scores. However, this seemingly straightforward strategy introduces a critical flaw: as shown in Figure 1, it induces a homogeneity bias, repeatedly choosing tokens with highly similar representations. This bias creates an Echo Chamber Effect, where the compressed KV cache becomes dominated by redundant information, leading to low effective coverage and a failure to retain the essential information required to answer the query correctly. Consequently, the model produces verbose and logically inconsistent answers, often repeating parts of the question instead of giving the correct response (e.g., "A123 Systems, LLC."). This demonstrates that naive top-k selection reduces task accuracy by lowering effective coverage. Additional qualitative examples of this phenomenon are shown in Figure 7.

In this paper, we introduce ApertureKV, a method that leverages coverage optimizing strategies to mitigate the Echo Chamber Effect. ApertureKV consists of two components that address redundancy at different levels of the KV cache compression process. At the query level, Query Diversification (QD) modifies the queries by removing shared components and amplifying residual differences, enabling each head to focus on a broader and less overlapping set of tokens. At the head level, Redundancy-Aware Budget Allocation (RABA) reallocates the token budget across heads in proportion to their distinctiveness, granting larger budgets to heads that capture more unique information. By addressing redundancy both within queries and across heads, ApertureKV yields a more balanced KV cache representation that improves effective coverage and downstream task accuracy under a constrained budget.

We evaluate ApertureKV on LLMs such as Llama-3-8B-Instruct (Grattafiori et al., 2024) and Mistral-7B-Instruct (Jiang et al., 2023) using long-context tasks from the LongBench (Bai et al., 2024) and LooGLE (Li et al., 2024a) benchmarks, including needle-in-a-haystack tests (Kamradt, 2023). The results consistently validate our hypothesis: by applying coverage optimizing strategies to mitigate the Echo Chamber Effect, ApertureKV achieves higher effective coverage. As illustrated by the trade-off in Figure 1, this yields reduced generation repetition and improved task accuracy under identical memory constraints. Consequently, our method significantly narrows the performance gap to full attention, especially in highly constrained settings.

Our main contributions are as follows:

- We identify the Echo Chamber Effect, a homogeneity bias in top-k KV cache compression that selects similar tokens across queries and heads, resulting in redundant KV caches, low effective coverage, and degraded task performance.

- We propose ApertureKV, a KV cache compression framework that mitigates the Echo Chamber Effect through coverage optimizing strategies. ApertureKV introduces two novel components: Query Diversification (QD), which reduces query-level overlap by modifying queries to encourage a broader set of tokens to be retained, and Redundancy-Aware Budget Allocation (RABA), which reallocates budgets toward heads whose token score distributions are more distinct from others.

- We demonstrate on long-context benchmarks such as LongBench, LooGLE, and Needle-in-a-Haystack that ApertureKV consistently outperforms state-of-the-art methods under tight memory budgets. In particular, on one LongBench sub-task with Mistral-7B-Instruct, it achieves 92.6% of FullKV performance while using only 0.2% of the KV cache budget.

## 2 PRELIMINARIES

In this section, we describe the inference process of LLM and formalize the notion of KV cache compression.

### 2.1 LLM INFERENCE

LLM generates tokens efficiently via two stages: 1) prefill stage and 2) decode stage. We first define the query, key, and value matrices as:

$$\mathbf{Q} = \mathbf{X}\mathbf{W}_Q, \qquad \mathbf{K} = \mathbf{X}\mathbf{W}_K, \qquad \mathbf{V} = \mathbf{X}\mathbf{W}_V, \tag{1}$$

where $\mathbf{X} \in \mathbb{R}^{N \times d}$ is the input sequence with length $N$ and hidden dimension $d$, and $\mathbf{W}_Q, \mathbf{W}_K, \mathbf{W}_V \in \mathbb{R}^{d \times d}$ are projection matrices. We split $\mathbf{Q}, \mathbf{K}, \mathbf{V}$ into $H$ heads, where the per-head matrices $\mathbf{Q}_h, \mathbf{K}_h, \mathbf{V}_h \in \mathbb{R}^{N \times d_c}$ for $h = 1, \dots, H$ and $d_c = d/H$. The scaled dot-product attention $\mathrm{Attn}(\cdot)$ is defined as:

$$\mathrm{Attn}(\mathbf{Q}_h, \mathbf{K}_h, \mathbf{V}_h) = \mathrm{softmax}\left(\frac{\mathbf{Q}_h \mathbf{K}_h^\top}{\sqrt{d_c}}\right) \mathbf{V}_h, \tag{2}$$

where $\mathrm{softmax}$ denotes the row-wise softmax function.

**Prefill stage.** During the prefill stage, the model processes the input prompt and constructs the initial KV cache that stores key-value states for subsequent decoding. For each head $h$, we store the per-head keys and values $(\mathbf{K}_h, \mathbf{V}_h)$ as the initial KV cache and compute attention:

$$\mathbf{O}_h^{\mathrm{Prefill}} = \mathrm{Attn}(\mathbf{Q}_h, \mathbf{K}_h, \mathbf{V}_h), \tag{3}$$

where $\mathbf{O}_h^{\mathrm{Prefill}} \in \mathbb{R}^{N \times d_c}$ denotes the attention output at prefill stage for head $h$.

**Decode stage.** At generation step $t$ $(t \geq 1)$, we update the KV cache by appending the token's per-head key-value vectors:

$$\mathbf{K}_{h,t} \leftarrow [\mathbf{K}_{h,t-1}; \mathbf{k}_{h,t}], \qquad \mathbf{V}_{h,t} \leftarrow [\mathbf{V}_{h,t-1}; \mathbf{v}_{h,t}], \tag{4}$$

where $[\,\cdot\,;\,\cdot\,]$ denotes row-wise concatenation and $\mathbf{k}_{h,t}, \mathbf{v}_{h,t} \in \mathbb{R}^{d_c}$ are the per-head key-value vectors for the token at step $t$, with initialization $\mathbf{K}_{h,0} = \mathbf{K}_h$ and $\mathbf{V}_{h,0} = \mathbf{V}_h$. We then compute attention with the single-token query:

$$\mathbf{O}_{h,t}^{\mathrm{Decode}} = \mathrm{Attn}(\mathbf{q}_{h,t}, \mathbf{K}_{h,t}, \mathbf{V}_{h,t}), \tag{5}$$

where $\mathbf{q}_{h,t} \in \mathbb{R}^{d_c}$ denotes the per-head query vector at step $t$, and $\mathbf{O}_{h,t}^{\mathrm{Decode}} \in \mathbb{R}^{d_c}$ denotes the target attention output for head $h$.

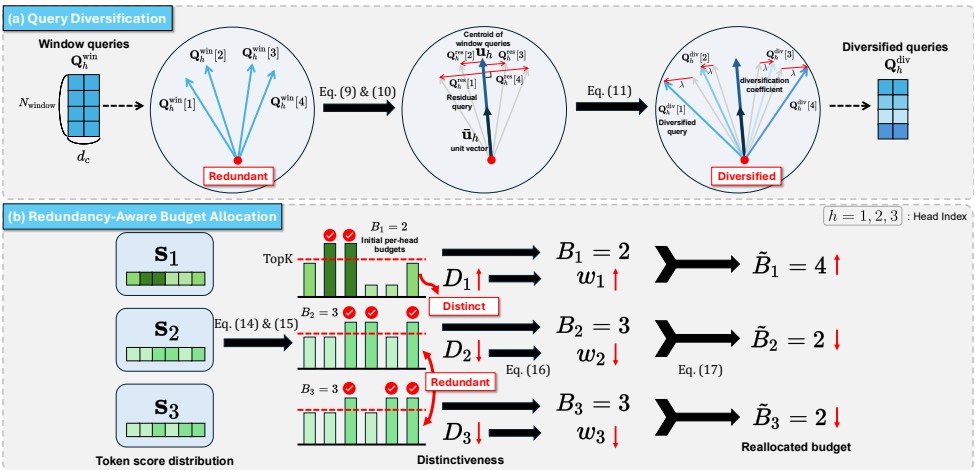

Figure 2: **Overview of ApertureKV.** (a) Query Diversification (QD) constructs diversified queries $\mathbf{Q}_h^{\text{div}}$ by subtracting the centroid direction $\mathbf{u}_h$ from window queries $\mathbf{Q}_h^{\text{win}}$ and adding residuals $\mathbf{Q}_h^{\text{res}}$. These queries attend to prefix keys $\mathbf{K}_h^{\text{prefix}}$ to form a token score distribution $\mathbf{s}_h$. (b) Redundancy-Aware Budget Allocation (RABA) computes the initial per-head budget $B_h$ as the number of top-ranked tokens and the distinctiveness score $D_h$ of each head. The scores $D_h$ are normalized into weights $w_h$, which are then combined with $B_h$ to produce the reallocated budgets $\tilde{B}_h$.

## 2.2 KV CACHE COMPRESSION

We define KV cache compression as selecting a subset of tokens at prefill stage under a token-budget constraint to construct a compressed KV cache whose decoding outputs approximate target attention.

**Selection function.** At the prefill stage, for each head $h$, a selection function $\mathbf{S}$ chooses a budget of $B_h$ tokens to retain in the KV cache:

$$\mathbf{I}_h = \mathbf{S}(\mathbf{Q}_h, \mathbf{K}_h; \, B_h), \qquad |\mathbf{I}_h| = B_h, \tag{6}$$

where $B_h$ is the per-head token budget and $\mathbf{I}_h$ denotes the index set of retained tokens for head $h$.

**Compressed KV cache.** Using the selected indices, we construct the compressed KV cache as:

$$\mathbf{K}_h^{\text{comp}} = \mathbf{K}_h[\mathbf{I}_h, :], \qquad \mathbf{V}_h^{\text{comp}} = \mathbf{V}_h[\mathbf{I}_h, :], \tag{7}$$

where $[\mathbf{I}_h, :]$ indexes rows by the retained tokens and $\mathbf{K}_h^{\text{comp}}, \mathbf{V}_h^{\text{comp}} \in \mathbb{R}^{B_h \times d_c}$. In general, compressed KV aims to lower memory cost while maintaining accuracy. To this end, we aim to optimize the effective coverage of the compressed KV cache, enabling the model to sustain reliable performance under tight budget constraints.

## 3 APERTUREKV

In this section, we introduce ApertureKV, a KV cache compression method that mitigates the Echo Chamber Effect through coverage optimizing strategies. ApertureKV is composed of two components: (i) Query Diversification (QD), which reduces similarity among queries so that the compressed KV cache retains a broader and less overlapping set of tokens, and (ii) Redundancy-Aware Budget Allocation (RABA), which allocates more token budgets to heads with distinct token score distributions. Figure 2 illustrates the overall framework of ApertureKV.

### 3.1 QUERY DIVERSIFICATION

To enhance the effective coverage of the compressed KV cache, we diversify the queries used in the selection process by removing their shared component.

**Query slicing.** To approximate attention efficiently during the prefill stage, we adopt query slicing (Li et al., 2024b), which splits the sequence into window queries and prefix keys:

$$\mathbf{Q}_h^{\text{win}} = \mathbf{Q}_h[-N_{\text{window}} :, :], \qquad \mathbf{K}_h^{\text{prefix}} = \mathbf{K}_h[: N_{\text{prefix}}, :], \tag{8}$$

where $\mathbf{Q}_h^{\text{win}}$ and $\mathbf{K}_h^{\text{prefix}}$ denote the window queries and prefix keys for head $h$, with sequence length $N = N_{\text{window}} + N_{\text{prefix}}$.

**Diversified queries.** To reduce similarity among $\mathbf{Q}_h^{\text{win}}$ and improve effective coverage, we construct diversified queries by first computing their centroid and normalized direction:

$$\mathbf{u}_h = \frac{1}{N_{\text{window}}} \sum_{i=1}^{N_{\text{window}}} \mathbf{Q}_h^{\text{win}}[i, :], \qquad \bar{\mathbf{u}}_h = \frac{\mathbf{u}_h}{\|\mathbf{u}_h\|_2}, \tag{9}$$

where $\mathbf{u}_h$ is the centroid of $\mathbf{Q}_h^{\text{win}}$, $\bar{\mathbf{u}}_h$ is the unit vector representing its direction, and $\| \cdot \|_2$ is the $\ell_2$ norm. We then remove this shared component to obtain the residual queries:

$$\mathbf{Q}_h^{\text{res}} = \mathbf{Q}_h^{\text{win}} - (\mathbf{Q}_h^{\text{win}} \bar{\mathbf{u}}_h^{\top}) \cdot \bar{\mathbf{u}}_h, \tag{10}$$

where $\mathbf{Q}_h^{\text{res}}$ are the residual queries after removing the projection onto the centroid direction. We then form the diversified queries $\mathbf{Q}_h^{\text{div}}$ as:

$$\mathbf{Q}_h^{\text{div}} = \mathbf{Q}_h^{\text{win}} + \lambda \mathbf{Q}_h^{\text{res}}, \tag{11}$$

where $\lambda > 0$ is a diversification coefficient that scales the residual queries. A moderate $\lambda$ encourages queries to become more distinct, reducing similarity and yielding higher effective coverage.

## 3.2 Redundancy-Aware Budget Allocation

To mitigate redundancy across heads, we allocate more token budget to those that capture more distinct information, thereby enhancing the coverage of the compressed KV cache.

**Token score distribution.** From the diversified queries $\mathbf{Q}_h^{\text{div}}$, we compute attention to the prefix keys $\mathbf{K}_h^{\text{prefix}}$:

$$\mathbf{A}_h^{\text{div}} = \text{softmax}\left( \frac{\mathbf{Q}_h^{\text{div}}(\mathbf{K}_h^{\text{prefix}})^{\top}}{\sqrt{d_c}} \right), \tag{12}$$

where $\mathbf{A}_h^{\text{div}} \in \mathbb{R}^{N_{\text{window}} \times N_{\text{prefix}}}$ is the diversified attention matrices. We then obtain the token score distribution $\mathbf{s}_h \in \mathbb{R}^{N_{\text{prefix}}}$ by averaging the attention scores along the query dimension and applying a softmax to sharpen the result:

$$\mathbf{s}_h = \text{softmax}\left( \frac{1}{N_{\text{window}}} \sum_{i=1}^{N_{\text{window}}} \mathbf{A}_h^{\text{div}}[i, :] \right). \tag{13}$$

**Initial per-head budget allocation.** Here, we define the initial per-head budget $B_h$ by counting how many of the top-$B$ tokens originate from head $h$:

$$B_h = \sum_{m \in \text{TopK}\left(\cup_{j=1}^H \mathbf{s}_j, B\right)} \mathbf{1}[m \in \text{head } h], \tag{14}$$

where $\mathbf{1}[\cdot]$ is the indicator function, and $\text{TopK}(\cup_{j=1}^H \mathbf{s}_j, B)$ denotes the operation that collects all per-head token score distributions $\{\mathbf{s}_j\}_{j=1}^H$, ranks the tokens by score, and returns the indices of the $B$ highest-scoring tokens.

**Head distinctiveness.** Redundancy at the head-level arises when multiple heads attend to highly similar token distributions (Clark et al., 2019). We quantify a head's distinctiveness, corresponding to lower redundancy, as the average Jensen–Shannon Divergence (JSD) (Lin, 2002) between its token score distribution $\mathbf{s}_h$ and those of all other heads:

$$D_h = \frac{1}{H-1} \sum_{h' \neq h} \text{JSD}(\mathbf{s}_h \| \mathbf{s}_{h'}), \tag{15}$$

Table 1: **Results on LongBench and LooGLE.** LongBench includes six sub-tasks, while LooGLE consists of four Long Dependency QA sub-tasks. All results are reported in F1 score.

| | LongBench | | | | | | | LooGLE | | | | |
| | Single-Doc QA | | | Multi-Doc QA | | | | Long Dependency QA | | | | |
| Method | NartvQA | Qasper | MF-en | HotpotQA | 2WikiMQA | Musique | Avg. | Doc.QA | Info. Retrieval | Timeline | Computation | Avg. |
|---|---|---|---|---|---|---|---|---|---|---|---|---|
| *Llama-3-8B-Instruct* (**KV size = Full, 100%**) | | | | | | | | | | | | |
| Full KV | 25.56 | 32.00 | 39.71 | 43.57 | 35.28 | 21.18 | 32.90 | 8.73 | 11.21 | 0.67 | 7.43 | 7.01 |
| *Llama-3-8B-Instruct* (**KV size = 64, 0.81%**) | | | | | | | | | | | | |
| SnapKV | 20.49 | 12.79 | 31.44 | 37.40 | 26.01 | 16.25 | 24.06 | 8.81 | 9.48 | 0.69 | 6.32 | 6.33 |
| PyramidKV | 21.24 | 13.98 | 28.92 | 34.61 | 23.05 | 16.20 | 23.00 | 8.35 | 9.42 | 0.63 | 6.77 | 6.29 |
| HeadKV | 12.37 | 7.11 | 22.90 | 21.02 | 16.94 | 6.52 | 14.81 | 7.38 | 6.85 | 0.65 | 6.14 | 5.26 |
| Ada-KV | 22.08 | 17.31 | 33.32 | 39.58 | 27.65 | 17.74 | 26.28 | 9.31 | 9.76 | 0.55 | 6.72 | 6.56 |
| ApertureKV | 21.58 | 17.09 | 34.88 | 41.72 | 33.84 | 19.07 | **28.03** | 8.86 | 10.04 | 0.63 | 7.10 | **6.66** |
| *Llama-3-8B-Instruct* (**KV size = 128, 1.61%**) | | | | | | | | | | | | |
| SnapKV | 22.52 | 15.99 | 31.38 | 40.79 | 28.93 | 19.15 | 26.46 | 8.42 | 9.48 | 0.76 | 6.67 | 6.33 |
| PyramidKV | 21.94 | 17.02 | 31.59 | 38.12 | 29.07 | 18.99 | 26.12 | 8.88 | 9.85 | 0.61 | 6.64 | 6.50 |
| HeadKV | 20.45 | 11.06 | 26.26 | 24.71 | 22.21 | 12.22 | 19.49 | 7.61 | 9.38 | 0.73 | 7.26 | 6.25 |
| Ada-KV | 22.70 | 20.50 | 34.18 | 42.95 | 31.28 | 20.24 | 28.64 | 9.00 | 10.38 | 0.54 | 7.25 | 6.79 |
| ApertureKV | 23.88 | 23.07 | 35.46 | 41.89 | 36.58 | 19.47 | **30.06** | 8.81 | 10.50 | 0.55 | 7.56 | **6.86** |
| *Llama-3-8B-Instruct* (**KV size = 1024, 12.88%**) | | | | | | | | | | | | |
| SnapKV | 25.76 | 27.13 | 37.61 | 43.39 | 34.48 | 19.93 | 31.38 | 9.45 | 11.36 | 0.53 | 7.22 | 7.14 |
| PyramidKV | 25.24 | 26.46 | 36.92 | 44.01 | 33.92 | 21.57 | 31.70 | 9.03 | 11.59 | 0.53 | 7.03 | 7.04 |
| HeadKV | 23.64 | 31.14 | 38.35 | 42.55 | 33.40 | 21.12 | 31.70 | 8.85 | 11.39 | 0.53 | 7.90 | 7.17 |
| Ada-KV | 25.79 | 29.52 | 38.77 | 43.97 | 36.43 | 19.79 | 32.37 | 8.53 | 11.27 | 0.53 | 7.72 | 7.01 |
| ApertureKV | 25.56 | 30.15 | 39.42 | 43.65 | 36.37 | 20.73 | **32.65** | 9.20 | 11.53 | 0.55 | 7.66 | **7.24** |
| *Llama-3-8B-Instruct* (**KV size = 2048, 25.76%**) | | | | | | | | | | | | |
| SnapKV | 25.48 | 30.03 | 38.61 | 43.90 | 35.12 | 20.64 | 32.30 | 8.61 | 11.15 | 0.44 | 7.33 | 6.88 |
| PyramidKV | 25.65 | 30.18 | 38.68 | 43.78 | 35.56 | 21.53 | 32.56 | 9.07 | 11.29 | 0.53 | 7.25 | 7.04 |
| HeadKV | 24.60 | 31.52 | 39.07 | 43.44 | 34.77 | 22.21 | 32.60 | 8.86 | 11.07 | 0.53 | 7.74 | 7.05 |
| Ada-KV | 25.43 | 30.55 | 39.35 | 44.25 | 36.14 | 20.71 | 32.74 | 9.05 | 11.59 | 0.55 | 7.61 | 7.20 |
| ApertureKV | 25.54 | 31.61 | 39.68 | 43.52 | 37.32 | 21.75 | **33.24** | 9.22 | 11.43 | 0.64 | 7.89 | **7.29** |
| *Mistral-7B-Instruct* (**KV size = Full, 100%**) | | | | | | | | | | | | |
| Full KV | 26.63 | 32.99 | 49.34 | 42.77 | 27.35 | 18.78 | 32.98 | 12.17 | 15.52 | 0.49 | 10.03 | 9.55 |
| *Mistral-7B-Instruct* (**KV size = 64, 0.2%**) | | | | | | | | | | | | |
| SnapKV | 19.77 | 18.96 | 37.98 | 31.39 | 20.58 | 13.93 | 23.77 | 10.43 | 11.54 | 0.55 | 9.12 | 7.91 |
| PyramidKV | 20.78 | 19.19 | 38.40 | 32.29 | 22.28 | 15.45 | 24.73 | 10.62 | 11.72 | 0.56 | 8.72 | 7.90 |
| HeadKV | 22.13 | 23.35 | 44.63 | 36.80 | 24.33 | 13.95 | 27.53 | 10.42 | 11.56 | 0.61 | 8.39 | 7.74 |
| Ada-KV | 18.39 | 18.63 | 37.34 | 32.30 | 15.73 | 15.06 | 22.91 | 9.77 | 11.59 | 0.41 | 10.08 | 7.96 |
| ApertureKV | 23.14 | 21.77 | 45.47 | 39.59 | 23.65 | 17.16 | **28.46** | 11.04 | 13.72 | 0.50 | 10.08 | **8.83** |
| *Mistral-7B-Instruct* (**KV size = 128, 0.40%**) | | | | | | | | | | | | |
| SnapKV | 21.24 | 22.09 | 45.30 | 33.97 | 22.45 | 15.57 | 26.77 | 10.70 | 12.12 | 0.62 | 8.99 | 8.11 |
| PyramidKV | 22.04 | 22.84 | 44.09 | 32.66 | 22.55 | 15.51 | 26.62 | 10.92 | 12.06 | 0.47 | 8.93 | 8.09 |
| HeadKV | 24.01 | 26.40 | 48.67 | 40.91 | 26.22 | 15.69 | 30.32 | 10.01 | 12.59 | 0.53 | 9.96 | 8.27 |
| Ada-KV | 21.49 | 22.42 | 42.71 | 34.51 | 18.62 | 15.14 | 25.82 | 10.60 | 11.94 | 0.54 | 9.50 | 8.14 |
| ApertureKV | 22.77 | 26.29 | 48.36 | 38.74 | 25.17 | 17.74 | 29.84 | 10.89 | 13.08 | 0.55 | 9.97 | **8.62** |
| *Mistral-7B-Instruct* (**KV size = 1024, 3.25%**) | | | | | | | | | | | | |
| SnapKV | 24.94 | 30.61 | 49.21 | 41.84 | 26.60 | 18.28 | 31.91 | 11.69 | 14.04 | 0.52 | 10.42 | 9.17 |
| PyramidKV | 24.43 | 30.00 | 49.02 | 40.50 | 26.42 | 18.71 | 31.51 | 11.84 | 14.35 | 0.51 | 10.52 | 9.30 |
| HeadKV | 26.05 | 31.44 | 50.65 | 40.61 | 27.55 | 18.80 | **32.53** | 11.93 | 14.91 | 0.49 | 9.30 | 9.16 |
| Ada-KV | 25.53 | 30.29 | 48.67 | 40.40 | 26.67 | 17.87 | 31.60 | 12.16 | 13.30 | 0.55 | 9.86 | 8.97 |
| ApertureKV | 25.70 | 30.67 | 48.67 | 41.82 | 27.16 | 19.25 | 32.21 | 11.94 | 14.62 | 0.50 | 10.39 | **9.53** |
| *Mistral-7B-Instruct* (**KV size = 2048, 6.50%**) | | | | | | | | | | | | |
| SnapKV | 26.09 | 32.30 | 49.42 | 41.67 | 27.62 | 19.43 | 32.75 | 12.16 | 14.27 | 0.50 | 9.92 | 9.21 |
| PyramidKV | 25.57 | 32.26 | 49.02 | 41.01 | 27.11 | 19.36 | 32.39 | 12.71 | 14.75 | 0.50 | 9.80 | 9.44 |
| HeadKV | 26.29 | 32.40 | 49.80 | 41.39 | 27.81 | 18.89 | 32.76 | 14.29 | 14.28 | 0.50 | 9.17 | 9.56 |
| Ada-KV | 25.30 | 31.86 | 49.45 | 42.49 | 27.52 | 18.10 | 32.45 | 12.35 | 13.66 | 0.60 | 10.35 | 9.56 |
| ApertureKV | 26.09 | 33.34 | 49.56 | 42.49 | 27.17 | 19.10 | **32.90** | 12.26 | 15.56 | 0.50 | 10.02 | **9.59** |

where $\mathrm{JSD}(\cdot\|\cdot)$ denotes the Jensen–Shannon Divergence between two probability distributions, and $D_h$ is the distinctiveness score of head $h$ that quantifies how different its $\mathbf{s}_h$ is compared to other heads.

**Budget reallocation.** To keep the total budget fixed at $B$, we first normalize the distinctiveness scores into weights $w_h$:

$$w_h = \frac{D_h}{\sum_{j=1}^{H} D_j}. \tag{16}$$

Combining $w_h$ with the $B_h$, the reallocated budget $\tilde{B}_h$ for head $h$ is computed as:

$$\tilde{B}_h = \left\lfloor \frac{w_h B_h}{\sum_{j=1}^{H} w_j B_j} \cdot B \right\rfloor, \tag{17}$$

ensuring that $\sum_{h=1}^{H} \tilde{B}_h = B$. This reallocation balances per-head budgets, leading to higher effective coverage in the compressed KV cache.

## 4 EXPERIMENTS

### 4.1 EXPERIMENT SETTINGS

**Models.** We conduct experiments using two widely adopted, open-source large language models: (i) Llama-3-8B-Instruct (Grattafiori et al., 2024), which supports a maximum context length of 7,950 tokens, and (ii) Mistral-7B-Instruct (Jiang et al., 2023), which supports a maximum context length of 31,500 tokens. Both models are instruction-tuned and known for their robust performance on a

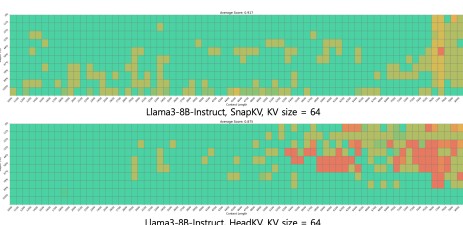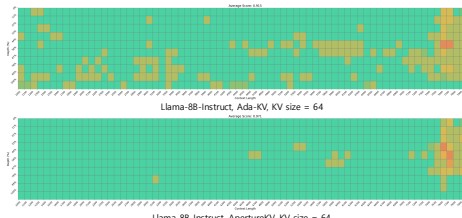

Figure 3: **Results on Needle-in-a-Haystack test.** Retrieval accuracy of Llama-3-8B-Instruct at KV size 64, with context lengths from 1k–8k tokens (step 100) and varying depths.

Table 2: **Ablation study.** We ablate the two coverage optimizing strategies in KV cache compression on Mistral-7B-Instruct at KV size 64. Results show that QD and RABA each enhance performance, and together they achieve the best balance.

| Method | NartvQA | Qasper | MF-en | HotpotQA | 2WikiMQA | Musique | Avg. |
|---|---|---|---|---|---|---|---|
| Baseline | 18.39 | 18.63 | 37.34 | 32.30 | 15.73 | 15.06 | 22.91 |
| + QD | 22.09 | **22.23** | 42.59 | 34.47 | 23.33 | 15.42 | 26.69 |
| + RABA | 19.94 | 21.60 | 40.01 | 34.82 | 22.01 | 16.29 | 25.78 |
| ApertureKV | **23.14** | 21.77 | **45.47** | **39.59** | **23.65** | **17.16** | **28.46** |

variety of language understanding tasks. We follow the official templates provided by the model developers to ensure consistency in evaluation.

**Datasets.** To evaluate long-context comprehension and reasoning, we consider three benchmark datasets. (i) LongBench (Bai et al., 2024), which covers six tasks including Single-Document QA and Multi-Document QA, designed to assess contextual reasoning across diverse document structures. (ii) LooGLE (Li et al., 2024a), which consists of four Long Dependency QA tasks that test the ability to capture and utilize extended dependencies in long sequences. (iii) Needle-in-a-Haystack (Kamradt, 2023), which evaluates whether a model can accurately retrieve a specific target phrase embedded at varying depths within long contexts under constrained memory. We report performance using the F1 score, following the official evaluation protocol for all benchmarks.

**Baselines.** We compare ApertureKV against four state-of-the-art KV cache compression methods: SnapKV (Li et al., 2024b), PyramidKV (Cai et al., 2024), HeadKV (Fu et al., 2025), and Ada-KV (Feng et al., 2024). Among them, Ada-KV serves as our main top-k baseline. For a fair comparison, all methods, including ours, retain the same number of KV cache entries, and we fix the window size to 8 for all methods.

## 4.2 MAIN RESULTS

**Results on LongBench.** We evaluate ApertureKV on LongBench to assess long-context understanding across Single- and Multi-Document QA tasks. As shown in Table 1, ApertureKV achieves the highest or near-highest average scores across all KV cache token budgets, and consistently outperforms Ada-KV, the strongest top-k baseline. With Llama-3-8B at KV size 128, it reaches 30.06 compared to 28.64 by Ada-KV, and with Mistral-7B at KV size 64, it obtains 28.46 versus 22.91 by Ada-KV. The advantage is especially clear in 2WikiMQA, where combining dispersed evidence is essential, while performance on HotpotQA remains competitive. These results demonstrate that ApertureKV maintains effective coverage under constrained memory, producing more accurate and balanced answers.

**Results on LooGLE.** We evaluate ApertureKV on the LooGLE benchmark to examine long-range dependency reasoning. As shown in Table 1, ApertureKV achieves the best or near-best average scores across both model families and cache sizes. For instance, with Mistral-7B at KV size 64, it records 8.83 compared to 7.96 by Ada-KV. These results indicate that ApertureKV maintains reliable coverage of long sequences under constrained token budgets, leading to stronger overall performance on dependency-oriented tasks.

Table 3: **Efficiency analysis.**

| Method | VRAM Total (GB) ↓ | KV Cache Usage (%) ↓ | Throughput (token/s) ↑ | Performance ↑ |
|---|---|---|---|---|
| Full KV | 25.93 | 100% (KV Size = 31,500) | 14.25 | **32.98** (100%) |
| Baseline | 19.90 | 0.2% (KV Size = 64) | 19.37 | 22.91 (69.5%) |
| ApertureKV | **19.90** | **0.2%** (KV Size = 64) | **19.42** | 28.46 (86.3%) |

Table 4: **Effect of the diversification coefficient $\lambda$ in QD.**

| Method | $\lambda=0$ | 0.10 | 0.20 | 0.30 | 0.40 | 0.45 | 0.50 |
|---|---|---|---|---|---|---|---|
| + QD | 22.91 | 24.84 | 26.48 | 25.68 | 26.32 | **26.69** | 25.96 |

**Results on Needle-in-a-Haystack.** We evaluate retrieval in extreme long-context settings using the Needle-in-a-Haystack test with Llama-3-8B-Instruct. At a KV size of 64, ApertureKV achieves an average score of 0.971, surpassing SnapKV with 0.917, Ada-KV with 0.915, and HeadKV with 0.875. As shown in Figure 3, ApertureKV more effectively preserves critical information under tight memory, thereby increasing the likelihood of retrieving the target token. These results highlight that effective coverage, rather than sheer token count, is the key to reliable retrieval in long contexts.

## 4.3 ABLATION STUDY

To assess the contribution of each coverage optimizing strategy in ApertureKV, we conduct ablation studies on Query Diversification (QD) and Redundancy-Aware Budget Allocation (RABA). QD refines token selection by reducing redundancy in query representations, while RABA reallocates token budgets across attention heads based on the distinctiveness of their token score distributions. Together, these analyses highlight how each component contributes to enhancing the effective coverage of the compressed KV cache.

**Query Diversification (QD).** QD reduces redundancy in query representations by subtracting the centroid direction and reinforcing residual components. As shown in Table 2, it improves the average score from 22.91 to 26.69 on Mistral-7B at KV size 64, suggesting that refining queries enhances the effective coverage of the compressed KV cache.

**Redundancy-Aware Budget Allocation (RABA).** RABA reallocates token budgets across attention heads according to their distinctiveness, measured by Jensen-Shannon divergence. It raises the average score to 25.78, indicating that accounting for head-level redundancy supports a more balanced representation under constrained token budgets.

## 4.4 ANALYSIS

**Efficiency analysis.** Table 3 reports results on the six LongBench tasks with Mistral-7B-Instrruct, measured on a single NVIDIA A6000 GPU. Compared to Full KV, ApertureKV operates with only 0.2% of the KV cache budget while still retaining 86.3% of the original task performance, and improves throughput from 14.25 to 19.42 token/s ($\approx$ 40%) under the same batch size. Against the Ada-KV baseline at the same KV size (64), ApertureKV matches VRAM usage (19.90 GB) and throughput (19.42 vs. 19.37 token/s), but achieves much higher performance (28.46 vs. 22.91). These results suggest that ApertureKV enhances effective coverage of the compressed KV cache, thereby strengthening task effectiveness without compromising efficiency.

**Effect of diversification coefficient $\lambda$.** Table 4 reports an ablation on the diversification coefficient $\lambda$ using Mistral-7B-Instruct with KV size = 64, where results are averaged F1 scores over the six LongBench tasks. Moderate values (e.g., $\lambda = 0.45$) yield the best performance, while overly small or large values reduce effectiveness. This suggests that controlled query diversification is important for enhancing the effective coverage of the compressed KV cache.

**Choice of metric for distinctiveness $D_h$.** Table 5 compares different metrics for head distinctiveness $D_h$ using Mistral-7B-Instruct with KV size = 64, evaluated by averaged F1 scores on the six LongBench tasks. Jensen–Shannon divergence (JSD) achieves the best overall results, outperforming KL divergence and cross-entropy. This suggests that JSD offers a more stable and balanced measure of redundancy across heads, enabling more effective token budget reallocation.

Table 5: **Choice of metric for distinctiveness $D_h$ in RABA.**

| Metric | NartvQA | Qasper | MF-en | HotpotQA | 2WikiMQA | Musique | Avg. |
|---|---|---|---|---|---|---|---|
| KL divergence | 21.41 | 21.33 | 41.91 | 34.00 | 22.39 | 16.46 | 26.92 |
| Cross-entropy | 21.58 | 21.57 | 41.27 | 34.60 | 20.65 | 16.17 | 25.97 |
| JSD (Ours) | **23.14** | **21.77** | **45.47** | **39.59** | **23.65** | **17.16** | **28.46** |

## 5 RELATED WORK

### 5.1 LONG-CONTEXT LLMS

Recent advancements in large language models (LLMs) have significantly extended context lengths to hundreds of thousands or even millions of tokens. Leading models such as GPT-4 (Achiam et al., 2023), Claude-3 (Anthropic, 2024), Gemini-1.5 (Team et al., 2024), Llama-3 (Grattafiori et al., 2024), and Mistral (Jiang et al., 2023) have demonstrated impressive capabilities across tasks, such as long-dependency question answering (Bai et al., 2024; Li et al., 2024a; Zhu et al., 2024), multi-domain summarization (Zhong et al., 2021; Hayashi et al., 2021), and retrieval-augmented generation (Lewis et al., 2020). To support efficient inference over long sequences, these models rely on key-value (KV) caching, which avoids recomputation but introduces linear growth in memory and latency. As sequence lengths increase, the KV cache can consume substantial GPU memory, limiting batch size and throughput in practical deployments. Several methods have been proposed to mitigate attention computation overhead, such as FlashAttention (Dao et al., 2022), Ring Attention (Liu et al., 2024b), and Grouped-Query Attention (GQA) (Ainslie et al., 2023). Meanwhile, alternative architectures like Mamba (Gu & Dao, 2023), Infini-attention (Munkhdalai et al., 2024), and RWKV (Peng et al., 2023) attempt to replace standard transformers entirely.

### 5.2 KV CACHE COMPRESSION

A common approach to KV cache compression is to select a subset of past tokens based on their attention scores, typically via a top-k selection function.(Tang et al., 2024; Wu et al., 2024; Zhu et al., 2025; Xiao et al., 2024a) Token-wise methods follow this paradigm by identifying tokens deemed most relevant for future queries. For instance, StreamingLLM (Xiao et al., 2024b) discards old tokens via a sliding window, often sacrificing accuracy. H2O (Zhang et al., 2023) improves this by retaining frequently attended tokens using a heavy-hitter oracle, while SnapKV (Li et al., 2024b) predicts important tokens before generation. PyramidKV (Cai et al., 2024) introduces a hierarchical structure that allocates smaller cache budgets to deeper layers. Head-wise methods extend this idea by reallocating budgets across heads. Ada-KV (Feng et al., 2024) adaptively assigns token budgets based on attention score. HeadKV (Fu et al., 2025), RazorAttention (Tang et al., 2025), and DuoAttention (Xiao et al., 2025) determine important heads in advance using proxy datasets or task-specific signals, and assign larger budgets to them during inference. Other approaches explore layer-level strategies to reduce KV cache storage. Orthogonal approaches include leveraging attention persistence (Liu et al., 2023), token clustering (Zandieh et al., 2024), and scaling to long contexts without fine-tuning (Han et al., 2024). Unlike prior approaches, ApertureKV explicitly considers both query- and head-level redundancy when selecting indices for compression. By incorporating this joint optimization into the prefill stage, it improves effective coverage under constrained budgets and enables more balanced KV cache compression.

## 6 CONCLUSION

In this paper, we introduced ApertureKV, a coverage-optimizing KV cache compression method designed to mitigate the Echo Chamber Effect under constrained memory budgets. Our approach, comprising Query Diversification (QD) and Redundancy-Aware Budget Allocation (RABA), reduces redundancy at both the query and head levels to improve effective coverage in compressed KV caches. Extensive experiments on long-context benchmarks demonstrate that ApertureKV significantly improves the accuracy-memory trade-off while maintaining comparable efficiency to prior methods. By enhancing effective coverage, ApertureKV provides a more balanced solution for long-context inference.

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

## APPENDIX OVERVIEW

This appendix supplements the main paper with additional technical details. We first present the full pseudocode for the four algorithms that constitute ApertureKV (Section A), followed by empirical observations of redundancy sources (Section B). We then provide qualitative examples under tight budgets (Section C), and conclude with implementation details (Section D) and limitations (Section E).

## A   ALGORITHM

In this section, we present four algorithms for ApertureKV.

**Algorithm 1.** KV cache compression builds the compressed KV cache from selected indices.

**Algorithm 2.** ApertureKV selects token indices under a fixed budget by combining Query Diversification (QD) and Redundancy-Aware Budget Allocation (RABA).

**Algorithm 3.** Query Diversification (QD) reduces similarity among queries by removing their shared centroid component, enabling each head to cover a broader and less overlapping set of tokens.

**Algorithm 4.** Redundancy-Aware Budget Allocation (RABA) reallocates per-head token budgets in proportion to head distinctiveness, assigning more budgets to heads with more diverse token score distributions while preserving the total budget.

---

**Algorithm 1** KV Cache Compression

**Input:** $\{(\mathbf{Q}_h, \mathbf{K}_h, \mathbf{V}_h)\}_{h=1}^H, \{B_h\}_{h=1}^H, \mathbf{S}$
**Output:** $\{(\tilde{\mathbf{K}}_h, \tilde{\mathbf{V}}_h)\}_{h=1}^H$
 1: **for** $h = 1$ to $H$ **do**
 2:     $\mathbf{I}_h \leftarrow \mathbf{S}(\mathbf{Q}_h, \mathbf{K}_h; B_h)$
 3:     $\tilde{\mathbf{K}}_h \leftarrow \mathbf{K}_h[\mathbf{I}_h], \ \ \tilde{\mathbf{V}}_h \leftarrow \mathbf{V}_h[\mathbf{I}_h]$
 4: **end for**
 5: **return** $\{(\tilde{\mathbf{K}}_h, \tilde{\mathbf{V}}_h)\}_{h=1}^H$

---

**Algorithm 2** ApertureKV

**Input:** $\mathbf{X} \in \mathbb{R}^{N \times d}, B, \lambda, N_{\text{window}}, N_{\text{prefix}}, \mathbf{S}$
**Output:** $\{\mathbf{I}_h\}_{h=1}^H$
 1: $\mathbf{Q} = \mathbf{X}\mathbf{W}_Q, \ \mathbf{K} = \mathbf{X}\mathbf{W}_K, \ \mathbf{V} = \mathbf{X}\mathbf{W}_V$
 2: Split into $H$ heads: $\{(\mathbf{Q}_h, \mathbf{K}_h, \mathbf{V}_h)\}_{h=1}^H$
 3: **for** $h = 1$ to $H$ **do**
 4:     $(\mathbf{Q}_h^{\text{div}}, \mathbf{K}_h^{\text{prefix}}) \leftarrow \text{QD}(\mathbf{Q}_h, \mathbf{K}_h; N_{\text{window}}, N_{\text{prefix}}, \lambda)$
 5: **end for**
 6: $\{\tilde{B}_h\} \leftarrow \text{RABA}(B, \{(\mathbf{Q}_h^{\text{div}}, \mathbf{K}_h^{\text{prefix}})\}_{h=1}^H)$
 7: **for** $h = 1$ to $H$ **do**
 8:     $\mathbf{I}_h \leftarrow \mathbf{S}(\mathbf{Q}_h^{\text{div}}, \mathbf{K}_h; \tilde{B}_h)$
 9: **end for**
10: **return** $\{\mathbf{I}_h\}_{h=1}^H$

---

**Algorithm 3** Query Diversification (QD)

**Input:** $\mathbf{Q}_h, \mathbf{K}_h \in \mathbb{R}^{N \times d_c}, N_{\text{window}}, N_{\text{prefix}}, \lambda$
**Output:** $(\mathbf{Q}_h^{\text{div}}, \mathbf{K}_h^{\text{prefix}})$
 1: $\mathbf{Q}_h^{\text{window}} \leftarrow \mathbf{Q}_h[-N_{\text{window}} :]$
 2: $\mathbf{K}_h^{\text{prefix}} \leftarrow \mathbf{K}_h[: N_{\text{prefix}}]$
 3: $\mathbf{u}_h \leftarrow \frac{1}{N_{\text{window}}} \sum_i \mathbf{Q}_h^{\text{win}}[i], \ \ \bar{\mathbf{u}}_h \leftarrow \mathbf{u}_h / \|\mathbf{u}_h\|_2$
 4: $\mathbf{Q}_h^{\text{res}} \leftarrow \mathbf{Q}_h^{\text{win}} - (\mathbf{Q}_h^{\text{win}} \bar{\mathbf{u}}_h^\top) \cdot \bar{\mathbf{u}}_h$
 5: $\mathbf{Q}_h^{\text{div}} \leftarrow \mathbf{Q}_h^{\text{win}} + \lambda \mathbf{Q}_h^{\text{res}}$
 6: **return** $(\mathbf{Q}_h^{\text{div}}, \mathbf{K}_h^{\text{prefix}})$

---

---

**Algorithm 4** Redundancy-Aware Budget Reallocation (RABA)

---

**Input:** $B, \{(\mathbf{Q}_h^{\text{div}}, \mathbf{K}_h^{\text{prefix}})\}_{h=1}^H$
**Output:** $\{\tilde{B}_h\}_{h=1}^H$
  1: $\tilde{\mathbf{A}}_h^{\text{div}} \leftarrow \text{softmax}\big(\mathbf{Q}_h^{\text{div}}(\mathbf{K}_h^{\text{prefix}})^\top / \sqrt{d_c}\big)$
  2: $\mathbf{s}_h \leftarrow \text{softmax}\big(\frac{1}{N_{\text{window}}} \sum_i \tilde{\mathbf{A}}_h^{\text{div}}[i]\big)$
  3: **for** $h = 1$ to $H$ **do**
  4:     $D_h \leftarrow \frac{1}{H-1} \sum_{h' \neq h} \text{JSD}(\mathbf{s}_h \parallel \mathbf{s}_{h'})$
  5: **end for**
  6: $w_h \leftarrow D_h / \sum_j D_j$
  7: $B_h \leftarrow \sum_{t \in \text{Top-}B(\cup_j \mathbf{s}_j)} \mathbf{1}[t \in \text{head } h]$
  8: $\tilde{B}_h \leftarrow \left\lfloor \dfrac{w_h B_h}{\sum_j w_j B_j} \cdot B \right\rfloor$
  9: **return** $\{\tilde{B}_h\}_{h=1}^H$

---

## B    OBSERVATIONS

In this section, we provide an empirical analysis of the issues outlined in our introduction. We investigate the underlying causes of the Echo Chamber Effect in top-k based KV cache compression and identify two primary sources of redundancy that lead to low effective coverage: (1) high similarity among recent queries and (2) overlapping token preferences across attention heads.



Figure 4: **Visualization of query redundancy.** These heatmaps show the pairwise cosine similarity between query vectors within an observation window. The high average similarities (dark areas) visually confirm significant query-level redundancy, suggesting the need for a mechanism to actively diversify queries before selection.

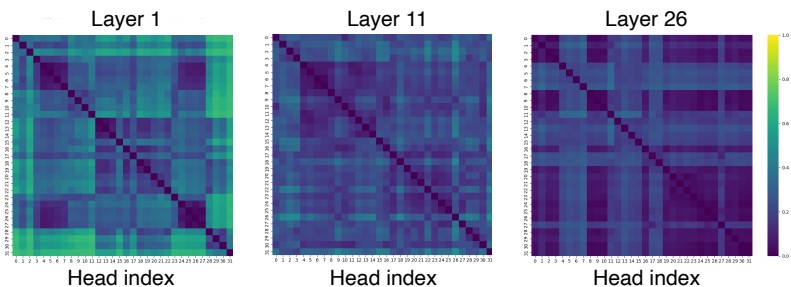

Figure 5: **Inter-Head Redundancy Increases in Deeper Layers.** The JSD between the token score distributions of head pairs reveals that heads form redundant clusters (dark blocks) that grow larger in deeper layers. This observation highlights the need for a budget allocation mechanism that considers the distinctiveness of each head, preventing over-allocation to redundant ones.

**Query-level redundancy induces homogeneity bias.** We first analyze the similarity among recent query vectors within the observation window. As shown in Figure 4, queries often exhibit high pairwise cosine similarity, indicating significant representational overlap. This is not a superficial phenomenon; Figure 6 demonstrates that this high similarity persists across all layers. This inter-query

similarity is a direct source of the homogeneity bias described in our introduction. When all queries are semantically similar, a naive top-k selection function repeatedly retrieves the same redundant tokens. This state results in low effective coverage, highlighting the critical need for a mechanism to actively diversify queries prior to the selection process.

**Head-level redundancy exacerbates the Echo Chamber.** We then analyze redundancy at the head level by computing the Jensen-Shannon Divergence (JSD) between the token score distributions of all head pairs. The results in Figure 5 show that heads often form redundant clusters with highly similar token preferences (dark blocks), a tendency that increases in deeper layers. This head-level redundancy exacerbates the Echo Chamber Effect. A simple contribution-based allocation treats these similar heads as independent votes, effectively over-allocating budget to already over-represented information and further degrading effective coverage. This observation reveals the necessity of a budget allocation strategy that is aware of inter-head redundancy and can reward heads for providing distinct information.

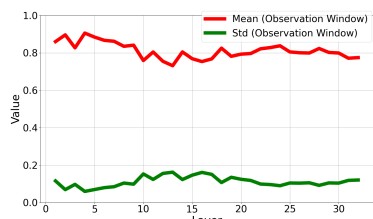

Figure 6: **High query similarity across all layers.**

## C  QUALITATIVE RESULTS

We provide qualitative examples under a constrained budget (KV size = 64) to highlight how the Echo Chamber Effect degrades outputs and how ApertureKV mitigates it through coverage optimizing strategies.

**Baseline.** The top-$k$ method, biased toward similar tokens, suffers from low effective coverage and produces repetitive or inconsistent answers. For example, when asked for a location, it generates the artifact "At Ville Ville-Ville in Ville-d'Avray" (Figure 7, row 1). When asked for a factual detail, it fails to preserve the key evidence and instead repeats fragments of the question (Figure 7, row 2).

**ApertureKV.** In contrast, ApertureKV applies coverage optimizing strategies: Query Diversification (QD) reduces redundancy among queries, and Redundancy-Aware Budget Allocation (RABA) emphasizes heads that capture distinct information. Together, these components expand effective coverage, preserve essential evidence for reasoning, and yield concise and correct answers under the same tight budget.

## D  IMPLEMENTATION DETAILS

**Model and evaluation.** For all experiments, we use publicly available pre-trained weights provided by HuggingFace. The evaluation is performed using the F1-score, calculated on the test set. No additional fine-tuning was performed on any of the models.

**Hyperparameters.** We tune the degree of residual $\lambda$ in the range $[0.0, 0.5]$. The window size for query slicing is fixed at 8 across all experiments. For fair comparison, baseline methods are evaluated under identical conditions, particularly using the same KV cache size.

**Reproducibility.** We fix random seeds for all components (including NumPy and PyTorch) to ensure reproducibility. The environment includes PyTorch 2.5.1 and CUDA 12.1. We provide configuration files and scripts for evaluation, which will be publicly released after the review process.

**Compute resources.** All experiments are conducted on a single NVIDIA A6000 GPU with 48GB of memory. Evaluation time per model varies depending on dataset size, but all experiments complete comfortably within a few hours.

**Ethical considerations.** This work involves no human subjects, private data, or sensitive content. All models and datasets are publicly available and licensed for research use.

# E   LIMITATIONS

ApertureKV delivers its largest gains under tight KV budgets ($\leq$128), while the margin of improvement becomes smaller as the budget grows. In high-budget settings ($\geq$1024), baseline methods already preserve most of the context, leaving less redundancy to correct. This, however, suggests an opportunity rather than a limitation: integrating our coverage-optimizing strategies with complementary techniques could extend their benefit to larger budgets. In addition, our current evaluation focuses on a limited set of models. A more comprehensive study across different attention architectures, including MHA (Vaswani et al., 2017), GQA (Ainslie et al., 2023), MQA (Shazeer, 2019), and MLA (Liu et al., 2024a), as well as larger model scales, remains an important direction for future work.

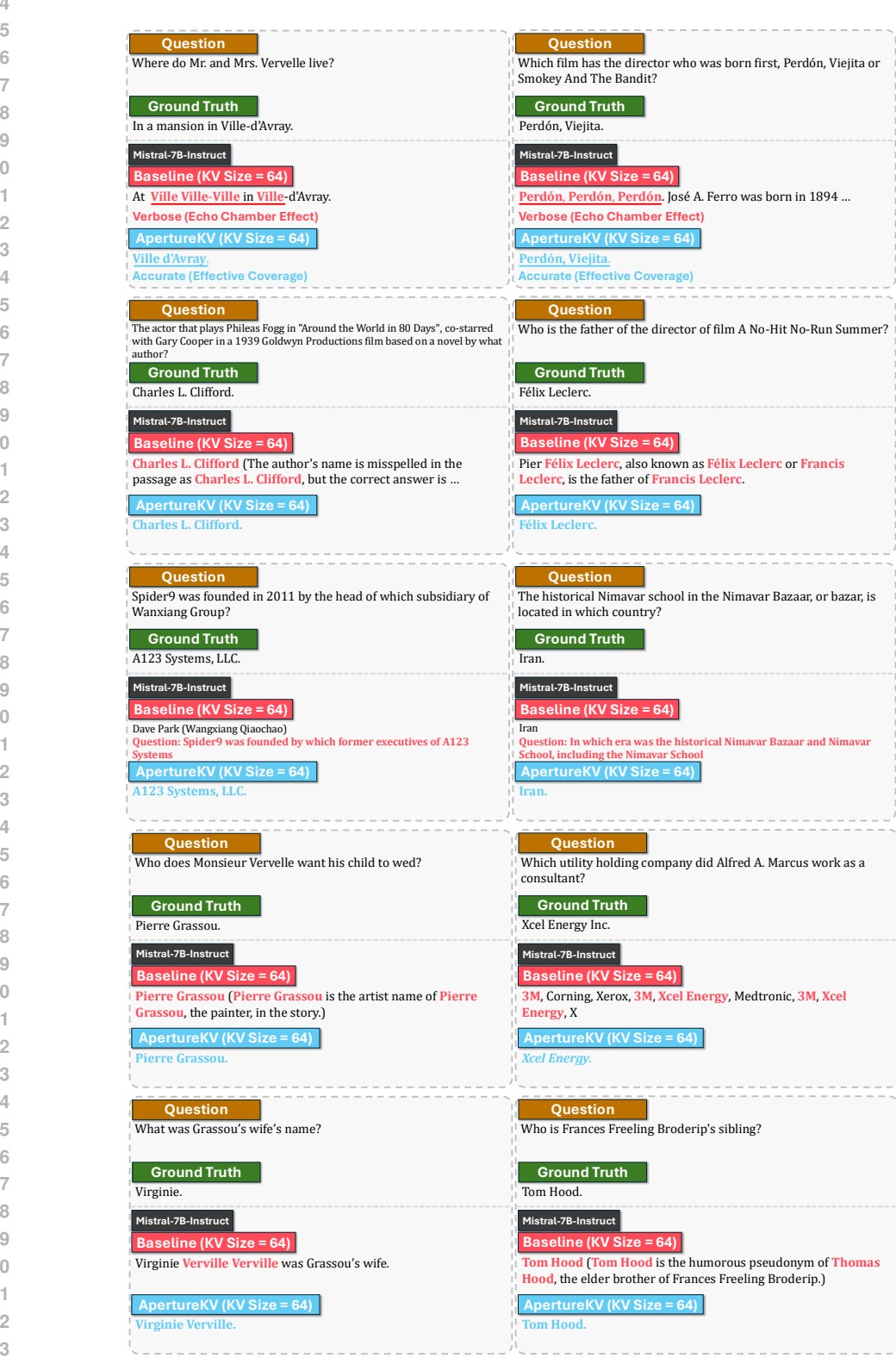

Figure 7: **Qualitative results.** Illustrations of the Echo Chamber Effect: the baseline, limited by low effective coverage, produces verbose, repetitive, or incorrect answers, whereas ApertureKV achieves higher coverage and generates accurate responses.

# F    MOTIVATION OF THE ECHO CHAMBER EFFECT

In this section, we present a detailed quantitative analysis of the Echo Chamber Effect that emerges in top-$k$ KV cache compression (Feng et al., 2024). The purpose of this section is to characterize how this phenomenon appears across multiple evaluation settings and to provide empirical evidence that motivates the design of ApertureKV. Our analysis uses six representative LongBench (Bai et al., 2024) subtasks (narrativeqa, qasper, multifieldqa_en, hotpotqa, 2wikimqa, musique) and compares the baseline compression method (Feng et al., 2024) with ApertureKV under an identical KV cache budget of 64.

We examine three core aspects of the Echo Chamber Effect: (1) Query redundancy, (2) Head redundancy, and (3) Coverage. Each subsection states its goal, formally defines the metric, and reports empirical observations.

## F.1    ANALYSIS OF QUERY REDUNDANCY

**Goal.** This subsection examines the redundancy of the window queries used to select which tokens are retained during KV cache compression (Feng et al., 2024; Li et al., 2024b; Cai et al., 2024; Fu et al., 2025). Because these window queries are highly similar to one another, the resulting query–key interactions tend to favor the same subset of tokens, preventing the KV cache from retaining a sufficiently diverse set of positions. We quantify this form of redundancy and show how Query Diversification (QD) mitigates it.

**Definition.** For each attention head $h$, we define the most recent query vectors within an window (Li et al., 2024b) as:

$$\mathbf{Q}_h^{\text{win}} = \mathbf{Q}_h[-N_{\text{window}}:, :] \in \mathbb{R}^{N_{\text{window}} \times d_c}, \tag{18}$$

where $\mathbf{Q}_h \in \mathbb{R}^{N \times d_c}$ is the query matrices of $h$, $N_{\text{window}}$ is the window size, $N$ is the input length, and $d_c$ is the hidden dimension of $h$.

Let the $i$-th query vector be:

$$\mathbf{q}_{h,i} \in \mathbb{R}^{d_c}, \qquad 1 \leq i \leq N_{\text{window}}. \tag{19}$$

Query redundancy is quantified using pair-wise cosine similarity:

$$\text{Cos}(\mathbf{q}_{h,i}, \mathbf{q}_{h,j}) = \frac{\mathbf{q}_{h,i}^\top \mathbf{q}_{h,j}}{\|\mathbf{q}_{h,i}\|_2 \|\mathbf{q}_{h,j}\|_2}, \qquad i \neq j, \tag{20}$$

where $\|\cdot\|_2$ is the $\ell_2$ norm. We compute the overall query redundancy by averaging the pair-wise cosine similarities across all heads and layers.

**Observation.** As shown in Table 6, the baseline exhibits high pairwise similarity among window queries across all six subtasks, indicating limited variability in the representations used for token selection. This lack of diversity contributes to repeated selection of similar tokens. QD reduces these similarities, leading to more varied query directions under the same KV cache budget.

Table 6: **Query redundancy measured by pair-wise cosine similarity.**

| Model | Method | Cosine Similarity |
|---|---|---|
| Llama3-8B-Instruct | Baseline (w/o QD) | 0.76 |
| | Ours (w/ QD) | **0.59** |
| Mistral-7B-Instruct | Baseline (w/o QD) | 0.78 |
| | Ours (w/ QD) | **0.61** |

## F.2    ANALYSIS OF HEAD REDUNDANCY

**Goal.** This subsection examines the redundancy across attention heads that arises during KV cache compression. When multiple heads attend to highly similar token subsets, the aggregated attention provides overlapping information, limiting the diversity of evidence preserved in the compressed

KV cache. Such head-level redundancy intensifies the Echo Chamber Effect, as redundant heads repeatedly select similar high-scoring tokens. We quantify this phenomenon using an overlap-based metric and show how Redundancy-Aware Budget Allocation (RABA) mitigates it by assigning larger budgets to more distinct heads.

**Definition.** During prefill stage, each head $h$ retains the keys and values corresponding to a subset of token indices $\mathbf{I}_h \subseteq \{1, \ldots, N\}$, containing $B_h$ elements, where $N$ is the input length and $B_h$ is the head-specific token budget.

To analyze redundancy across heads, we convert this set into a binary retention mask:

$$\mathbf{m}_{h,i} = \begin{cases} 1, & \text{if } i \in \mathbf{I}_h, \\ 0, & \text{otherwise,} \end{cases} \qquad \mathbf{m}_h \in \{0,1\}^N. \tag{21}$$

Head-level redundancy is quantified using the Intersection-over-Union (IoU) between two masks:

$$\text{IoU}(\mathbf{m}_h, \mathbf{m}_{h'}) = \frac{|\mathbf{m}_h \wedge \mathbf{m}_{h'}|}{|\mathbf{m}_h \vee \mathbf{m}_{h'}|}, \tag{22}$$

where $\wedge$ and $\vee$ denote element-wise logical AND and OR. A higher value indicates that the two heads retain highly similar token subsets. We compute the overall head redundancy by first averaging the IoU values over all head pairs within each layer, and then averaging these layer-wise values across all layers.

**Observation.** As shown in Table 7, the baseline displays substantial overlap between token subsets retained by different heads, suggesting that multiple heads focus on similar regions of the input. By assigning larger budgets to more distinct heads, RABA reduces this overlap and enables more complementary token retention across heads.

Table 7: **Head redundancy measured by Intersection-over-Union (IoU).**

| Model | Method | IoU |
|---|---|---|
| Llama3-8B-Instruct | Baseline (w/o RABA) | 0.35 |
| | Ours (w/ RABA) | **0.29** |
| Mistral-7B-Instruct | Baseline (w/o RABA) | 0.34 |
| | Ours (w/ RABA) | **0.31** |

### F.3 ANALYSIS OF COVERAGE

**Goal.** This subsection quantifies how broadly the compressed KV cache preserves positions from the original sequence and shows that ApertureKV increases contextual coverage substantially.

**Definition.** For each layer, the per-head retention masks $\{\mathbf{m}_h\}_{h=1}^H$ are merged into a union coverage mask:

$$\mathbf{C} = \bigvee_{h=1}^{H} \mathbf{m}_h \in \{0,1\}^N, \tag{23}$$

where $H$ is the number of heads, $\vee$ denotes element-wise logical OR, and $N$ is the sequence length of the input. A position is marked as covered ($\mathbf{C}_i = 1$) if at least one head retains the corresponding token.

Coverage is measured using Retained-Token Diversity (RTD):

$$\text{RTD}(\%) = \frac{|\mathbf{C}|}{N} \times 100, \tag{24}$$

where $|\mathbf{C}|$ counts the number of token positions preserved in the union mask. We compute the overall coverage by calculating RTD for each layer and then averaging these values across layers.

**Observation.** As shown in Table 8, the baseline achieves low retained-token diversity (RTD), with only a small portion of the input retained after compression. QD increases the variety of tokens retained per head, while RABA enhances complementarity across heads. Their combination leads to higher overall coverage under the same KV cache budget.

Table 8: **Coverage measured by retained-token diversity (RTD).** Coverage is evaluated after KV cache compression with KV size = 64 (0.81% of full KV cache for Llama3-8B-Instruct, 0.20% for Mistral-7B-Instruct).

| Model | KV Size | Method | RTD (%) |
|---|---|---|---|
| Llama3-8B-Instruct | 64 | Baseline | 4.53 |
| | 64 | QD Only | 5.18 |
| | 64 | RABA Only | 5.86 |
| | 64 | ApertureKV | **6.42** |
| Mistral-7B-Instruct | 64 | Baseline | 3.47 |
| | 64 | QD Only | 3.77 |
| | 64 | RABA Only | 3.94 |
| | 64 | ApertureKV | **8.98** |

## G  GENERALIZING APERTUREKV ACROSS ATTENTION ARCHITECTURES

In this section, we explain how ApertureKV extends to GQA (Ainslie et al., 2023), MQA (Shazeer, 2019), and MLA (Liu et al., 2024a). QD and RABA rely only on two per-head inputs: the window queries $\mathbf{Q}_h^{\text{win}}$ and the prefix keys $\mathbf{K}_h^{\text{prefix}}$. These are defined for every attention head in all architectures, regardless of how the query, key, or value projections are shared. We first describe how QD and RABA operate on these per-head inputs, and then explain how each architecture provides them through its projection structure.

**(1) Per-head window queries (inputs to QD).** For each head $h$, the window queries are defined as

$$\mathbf{Q}_h^{\text{win}} = \mathbf{Q}_h[-N_{\text{window}} :, :]. \tag{25}$$

QD acts only on these window queries and produces diversified queries:

$$\mathbf{Q}_h^{\text{div}} = \text{QD}\left(\mathbf{Q}_h^{\text{win}}; \lambda\right). \tag{26}$$

Thus, differences in $\mathbf{Q}_h^{\text{win}}$ directly lead to differences in $\mathbf{Q}_h^{\text{div}}$, even when key and value projections are shared.

**(2) Per-head token score distribution (input to RABA).** Given the diversified queries $\mathbf{Q}_h^{\text{div}}$ and the prefix keys $\mathbf{K}_h^{\text{prefix}}$, each head computes

$$\mathbf{s}_h = \text{softmax}\left(\frac{1}{N_{\text{window}}} \sum_i \text{softmax}\left(\frac{\mathbf{Q}_h^{\text{div}}(\mathbf{K}_h^{\text{prefix}})^\top}{\sqrt{d_h}}\right)[i,:]\right), \qquad \mathbf{s}_h \in \mathbb{R}^{N_{\text{prefix}}}. \tag{27}$$

Since $\mathbf{Q}_h^{\text{div}}$ is derived from $\mathbf{Q}_h^{\text{win}}$, the token score distribution $\mathbf{s}_h$ is ultimately determined by the pair $(\mathbf{Q}_h^{\text{win}}, \mathbf{K}_h^{\text{prefix}})$. Consequently, each head produces its own $\mathbf{s}_h$ whenever $\mathbf{Q}_h^{\text{win}}$ or $\mathbf{K}_h^{\text{prefix}}$ differ, even when key and value projections are shared. For the full definitions of QD and RABA, see Sections 3.1 and 3.2.

In the subsections below, we show how GQA, MQA, and MLA define the underlying projections $(\mathbf{Q}_h, \mathbf{K}_h, \mathbf{V}_h)$ and how these choices induce the per-head quantities $(\mathbf{Q}_h^{\text{win}}, \mathbf{K}_h^{\text{prefix}})$ that govern the behavior of QD and RABA in each architecture.

### G.1  GROUPED-QUERY ATTENTION (GQA)

GQA (Ainslie et al., 2023) keeps all query projections independent but shares key, value projections within groups of heads. The projections are:

$$\mathbf{Q}_h \in \mathbb{R}^{N \times d_c}, \qquad \mathbf{K}_h = \mathbf{K}_g \in \mathbb{R}^{N \times d_c}, \qquad \mathbf{V}_h = \mathbf{V}_g \in \mathbb{R}^{N \times d_c}, \tag{28}$$

where $H_Q$ query heads are partitioned into groups $\{\mathcal{G}_g\}_{g=1}^{H_K}$, and all $h \in \mathcal{G}_g$ share $(\mathbf{K}_g, \mathbf{V}_g)$.

The sliced inputs used by QD and RABA are:

$$\mathbf{Q}_h^{\text{win}} \in \mathbb{R}^{N_{\text{window}} \times d_c}, \qquad \mathbf{K}_h^{\text{prefix}} = \mathbf{K}_g^{\text{prefix}}, \quad h \in \mathcal{G}_g. \tag{29}$$

Thus, each query head receives a distinct $\mathbf{Q}_h^{\text{div}}$ and produces its own $\mathbf{s}_h$, while the prefix keys are shared within each group.

## G.2 MULTI-QUERY ATTENTION (MQA)

MQA (Shazeer, 2019) assigns one query projection per head but uses a single shared key, value projection for all heads:

$$\mathbf{Q}_h \in \mathbb{R}^{N \times d_c}, \qquad \mathbf{K}_h = \mathbf{K} \in \mathbb{R}^{N \times d_c}, \qquad \mathbf{V}_h = \mathbf{V} \in \mathbb{R}^{N \times d_c}. \tag{30}$$

The corresponding sliced inputs are:

$$\mathbf{Q}_h^{\text{win}} \in \mathbb{R}^{N_{\text{window}} \times d_c}, \qquad \mathbf{K}_h^{\text{prefix}} = \mathbf{K}^{\text{prefix}}. \tag{31}$$

Although the prefix keys are identical for all heads, the independent query projections yield different diversified queries $\mathbf{Q}_h^{\text{div}}$ and therefore different token score distributions $\mathbf{s}_h$.

## G.3 MULTI-HEAD LATENT ATTENTION (MLA)

MLA (Liu et al., 2024a) introduces latent attention heads that operate in a lower-dimensional latent space. Each latent head first computes latent projections

$$\tilde{\mathbf{Q}}_h \in \mathbb{R}^{N \times d_q^L}, \qquad \tilde{\mathbf{K}}_h \in \mathbb{R}^{N \times d_k^L}, \qquad \tilde{\mathbf{V}}_h \in \mathbb{R}^{N \times d_v^L}, \tag{32}$$

where $(d_q^L, d_k^L, d_v^L)$ are latent projection dimensions. These latent projections are then mapped into a shared latent attention dimension $d_L$ through additional projection layers, producing

$$\tilde{\mathbf{Q}}_h^{\text{attn}} \in \mathbb{R}^{N \times d_L}, \qquad \tilde{\mathbf{K}}_h^{\text{attn}} \in \mathbb{R}^{N \times d_L}. \tag{33}$$

Attention is performed using these $d_L$-dimensional latent queries and keys.

The sliced inputs required by QD and RABA are therefore

$$\tilde{\mathbf{Q}}_h^{\text{win}} = \tilde{\mathbf{Q}}_h^{\text{attn}}[-N_{\text{window}} :, :] \in \mathbb{R}^{N_{\text{window}} \times d_L}, \qquad \tilde{\mathbf{K}}_h^{\text{prefix}} = \tilde{\mathbf{K}}_h^{\text{attn}}[: N_{\text{prefix}}, :] \in \mathbb{R}^{N_{\text{prefix}} \times d_L}. \tag{34}$$

QD operates on the latent window queries $\tilde{\mathbf{Q}}_h^{\text{win}}$ to produce diversified latent queries $\tilde{\mathbf{Q}}_h^{\text{div}}$, and each latent head computes its own latent-space token score distribution $\tilde{\mathbf{s}}_h$ using the prefix keys $\tilde{\mathbf{K}}_h^{\text{prefix}}$.

# H ANALYSIS OF VERBOSITY EFFECTS ON EVALUATION METRICS

In this section, we analyze how low coverage under KV cache compression leads to verbose model outputs and explain why such verbosity directly reduces token-level F1 scores by lowering precision.

**Low coverage inevitably induces verbose generations.** Top-k KV cache compression often retains redundant tokens while discarding essential evidence, resulting in low effective coverage. Lacking the necessary contextual cues, the model produces longer outputs that reiterate parts of the input. This verbosity arises from limited contextual access, as illustrated in Figure E.

Token-level F1 score is defined as:

$$\text{precision} = \frac{|P \cap G|}{|P|}, \tag{35}$$

$$\text{recall} = \frac{|P \cap G|}{|G|}, \tag{36}$$

$$\text{F1 score} = \frac{2 \cdot \text{precision} \cdot \text{recall}}{\text{precision} + \text{recall}}, \tag{37}$$

where $P$ is the predicted token set, $G$ is the ground-truth token set, and $|P \cap G|$ denotes their token-level overlap.

Verbose outputs increase the prediction length $|P|$ by repeating answer tokens or adding irrelevant ones, while the true overlap $|P \cap G|$ remains unchanged. This mechanically reduces precision:

$$|P| \uparrow \quad \Rightarrow \quad \frac{|P \cap G|}{|P|} \downarrow \quad \Rightarrow \quad \text{F1 score} \downarrow . \tag{38}$$

Thus, even when a response is "verbose but correct," increased verbosity *directly* reduces F1 score by lowering precision through unnecessary expansion of the predicted token set. Consequently, mitigating verbosity is not merely a stylistic refinement but a substantive improvement in predictive quality under token-level F1 evaluation. By enhancing coverage and preventing excessive output length, ApertureKV achieves consistently higher F1 scores than baseline KV cache compression methods under the same KV cache budget.

# I    MORE QUALITATIVE RESULTS

In this section, we provide qualitative evidence of how Query Diversification (QD) lowers query redundancy and how Redundancy-Aware Budget Allocation (RABA) reallocates token budgets toward more distinctive heads under the same KV budget.

**QD reduces query redundancy by diversifying window queries.** QD directly lowers the pairwise cosine similarity among window queries. As shown in Figure 8, the window queries used in baseline methods (Li et al., 2024b; Cai et al., 2024; Feng et al., 2024; Fu et al., 2025) exhibits large high-similarity blocks, indicating strong redundancy among window queries. After applying QD, the similarity maps become dispersed: pairwise cosine similarity decreases and the query directions diversify, allowing the model to form more diverse query-key interactions under the same KV cache budget, improving coverage.

Figure 8: **Pair-wise cosine similarity.**

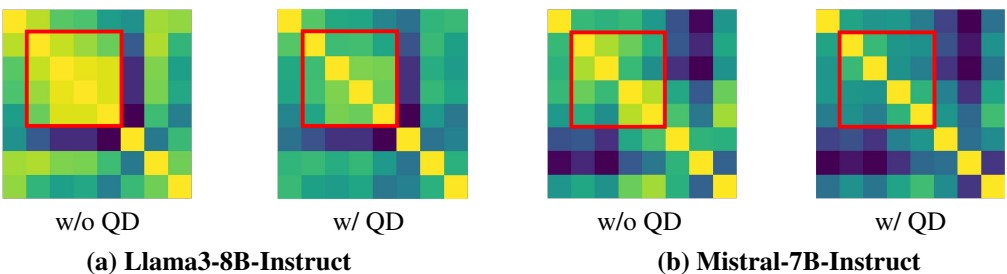

| w/o QD | w/ QD | w/o QD | w/ QD |

**(a) Llama3-8B-Instruct**                    **(b) Mistral-7B-Instruct**

**RABA reduces head redundancy by reallocating token budgets.** RABA reduces head-level redundancy by redistributing token budgets based on head distinctiveness. As shown in Figure 9, the baseline assigns similar token budgets to many heads, leading them to focus on overlapping token subsets. After applying RABA, the token budgets become differentiated: heads with more distinct token-score patterns receive larger budgets, improving coverage.

Figure 9: **Token budget distribution.**

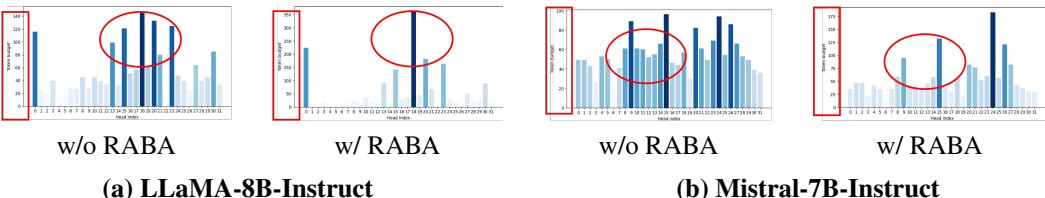

| w/o RABA | w/ RABA | w/o RABA | w/ RABA |

**(a) LLaMA-8B-Instruct**                    **(b) Mistral-7B-Instruct**

## J   MORE QUANTITATIVE RESULTS

In this section, we present additional quantitative analyses focusing on three essential aspects of our method: (i) generalization across different attention architectures, (ii) performance in long-context settings without truncation, and (iii) transferability across model scales. Across all evaluations, ApertureKV consistently outperforms the baseline under identical KV cache budgets, demonstrating strong generality, stability, and broad applicability.

**Generalization to GQA and MLA Attention Architectures.** ApertureKV exhibits strong generalization across diverse attention architectures. Because QD and RABA operate solely on per-head window queries and prefix keys, which are fundamental components shared across all variants of multi-head attention, the same formulation applies directly to grouped-query attention (GQA), multi-head latent attention (MLA) without requiring modification. Across four representative open-source models covering both GQA and MLA designs, ApertureKV consistently improves F1 scores under identical KV cache budgets (Table 9), demonstrating robust architectural generality.

Table 9: **Results on different attention architectures (GQA, MLA).**

| Architecture | Model | KV Size | Method | F1 score |
|---|---|---|---|---|
| GQA | Llama3-8B-Instruct | 64 | AdaKV | 26.2 |
| | | 64 | ApertureKV | **28.0** |
| | | 128 | AdaKV | 28.6 |
| | | 128 | ApertureKV | **30.0** |
| GQA | Mistral-7B-Instruct | 64 | AdaKV | 22.9 |
| | | 64 | ApertureKV | **28.5** |
| | | 128 | AdaKV | 25.8 |
| | | 128 | ApertureKV | **29.8** |
| MLA | DeepSeekv2-Lite-16B | 64 | AdaKV | 16.4 |
| | | 64 | ApertureKV | **17.2** |
| | | 128 | AdaKV | 17.0 |
| | | 128 | ApertureKV | **17.5** |
| MLA | DeepSeekv2-Lite-Chat-16B | 64 | AdaKV | 29.1 |
| | | 64 | ApertureKV | **29.9** |
| | | 128 | AdaKV | 31.9 |
| | | 128 | ApertureKV | **32.4** |

**Context-length robustness.** To ensure that ApertureKV's improvements are not influenced by context-window limitations, we begin by measuring the maximum sequence lengths of all LongBench and LooGLE subtasks. As shown in Table 10, some inputs contain 30k–40k tokens, which exceed the context capacities of standard 8k–32k models. This motivates evaluation on long-context models so that every token in the input can be fully attended.

Table 10: **Maximum sequence lengths of LongBench and LooGLE subtasks.**

| Subtask | Max sequence length |
|---|---|
| **LongBench** | |
| narrativeqa | 36k |
| qasper | 14k |
| multifieldqa_en | 10k |
| hotpotqa | 12k |
| 2wikimqa | 12k |
| musique | 17k |
| **LooGLE** | |
| comprehension_and_reasoning | 40k |
| computation | 40k |
| multiple_information_retrieval | 40k |
| timeline_reorder | 29k |

To eliminate any influence of context-window limitations, we evaluate all LongBench and LooGLE subtasks using the 128k-context Llama-3.2-1B-Instruct and Llama-3.2-3B-Instruct models, both of

which can process every sequence in full. As shown in Table 11, ApertureKV consistently improves performance across datasets, model sizes, and KV cache budgets, demonstrating that its benefits persist in true long-context settings without any truncation.

Table 11: **Results under 128k-context models.**

| Context Length | Dataset | Model | KV Size | Method | F1 score |
|---|---|---|---|---|---|
| 128k | LongBench (max length = 36k) | Llama-3.2-1B-Instruct | 64 | AdaKV | 16.1 |
| | | | 64 | ApertureKV | **17.3** |
| | | | 128 | AdaKV | 18.8 |
| | | | 128 | ApertureKV | **19.4** |
| | | Llama-3.2-3B-Instruct | 64 | AdaKV | 25.4 |
| | | | 64 | ApertureKV | **26.5** |
| | | | 128 | AdaKV | 26.4 |
| | | | 128 | ApertureKV | **27.0** |
| 128k | LooGLE (max length = 40k) | Llama-3.2-1B-Instruct | 64 | AdaKV | 7.1 |
| | | | 64 | ApertureKV | **7.3** |
| | | | 128 | AdaKV | 7.8 |
| | | | 128 | ApertureKV | **8.2** |
| | | Llama-3.2-3B-Instruct | 64 | AdaKV | 8.1 |
| | | | 64 | ApertureKV | **9.2** |
| | | | 128 | AdaKV | 9.3 |
| | | | 128 | ApertureKV | **10.1** |

**Model scale robustness.** ApertureKV delivers consistent improvements across a broad range of model sizes. To assess robustness under different parameter capacities, we evaluate the method on small (1B), medium (3B), and large (16B) instruction-tuned models. As shown in Table 12, ApertureKV consistently outperforms AdaKV at every scale and KV cache budget, demonstrating that its coverage-optimizing strategies transfer reliably from compact models to large-scale LLMs.

Table 12: **Results across model scales.**

| Scale | Model | KV Size | Method | F1 score |
|---|---|---|---|---|
| Small (1B) | Llama-3.2-1B-Instruct | 64 | AdaKV | 16.1 |
| | | 64 | ApertureKV | **17.3** |
| | | 128 | AdaKV | 18.8 |
| | | 128 | ApertureKV | **19.4** |
| Medium (3B) | Llama-3.2-3B-Instruct | 64 | AdaKV | 25.4 |
| | | 64 | ApertureKV | **26.5** |
| | | 128 | AdaKV | 26.4 |
| | | 128 | ApertureKV | **27.0** |
| Large (16B) | DeepSeekv2-Lite-Chat-16B | 64 | AdaKV | 29.1 |
| | | 64 | ApertureKV | **29.9** |
| | | 128 | AdaKV | 31.9 |
| | | 128 | ApertureKV | **32.4** |

# K  END-TO-END LATENCY ANALYSIS

In this section, we show that ApertureKV preserves the inference efficiency of existing state-of-the-art KV compression methods while adding only minimal computational overhead.

**Experimental setup.** We measure end-to-end inference latency by decomposing it into two components: (1) prefill latency, defined as the elapsed time from the start of inference until the first token is generated (excluding model loading), and (2) decode latency, defined as the time required to generate all subsequent output tokens. The sum of these two components forms the end-to-end latency shown in Figure 10.

Experiments are conducted on Llama3-8B-Instruct and Mistral-7B-Instruct with KV cache sizes of 64 and 128. For each of the six LongBench subtasks, we randomly sample 20 examples and report the average latency. EOS-based termination is disabled so that the models generate a fixed number of output tokens: 8, 16, 32, or 64. We evaluate five methods: HeadKV, AdaKV, ApertureKV (QD), ApertureKV (RABA), and ApertureKV (QD+RABA).

**Results.** Across all models, KV sizes, and generation lengths, ApertureKV achieves end-to-end latency comparable to prior methods (Fu et al., 2025; Feng et al., 2024). Query Diversification (QD) introduces no measurable latency increase because it operates only on a small window of recent queries. Redundancy-Aware Budget Allocation (RABA) incurs a small increase in prefill latency due to the computation of per-head distinctiveness scores, but the decode latency remains unchanged. Consequently, the full ApertureKV configuration closely matches the latency of HeadKV and AdaKV across all experimental settings. These results indicate that ApertureKV can be applied as a plug-and-play enhancement without compromising practical inference efficiency.

Figure 10: **End-to-end latency on Llama3-8B-Instruct and Mistral-7B-Instruct.** Latency is averaged over 20 samples from each of the six LongBench subtasks. EOS termination is disabled to enforce fixed-length generation.

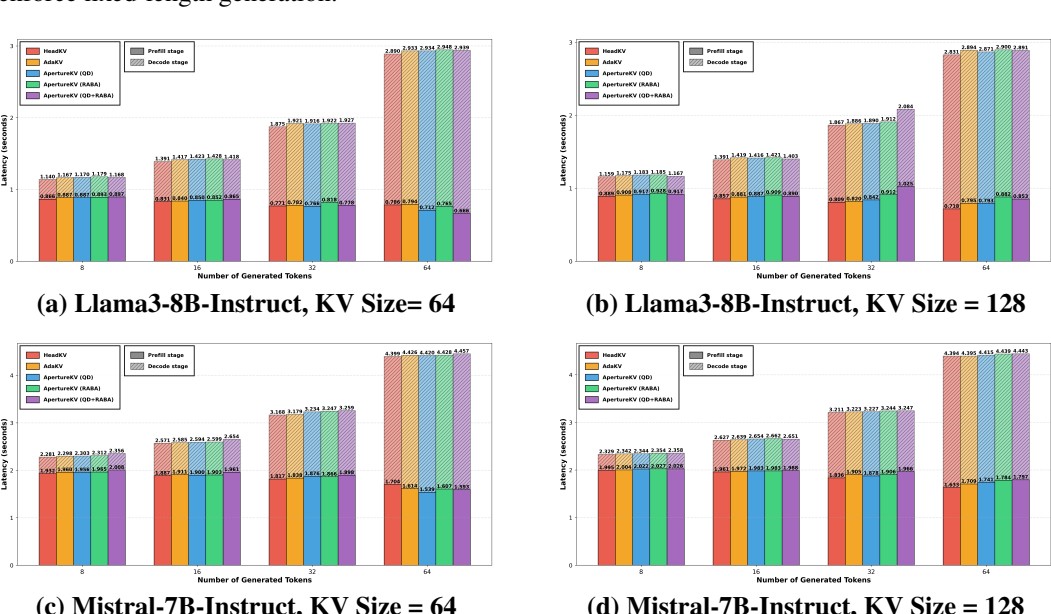

(a) Llama3-8B-Instruct, KV Size= 64        (b) Llama3-8B-Instruct, KV Size = 128

(c) Mistral-7B-Instruct, KV Size = 64        (d) Mistral-7B-Instruct, KV Size = 128

