# OpenReview forum: "Mitigating the Echo Chamber Effect in KV Cache Compression via Coverage Optimization"
_ICLR.cc/2026/Conference — ICLR 2026 Conference Desk Rejected Submission_

### Official Review · Reviewer_ucRM · 2025-10-29

**Soundness:** 3
**Presentation:** 3
**Contribution:** 3
**Rating:** 6
**Confidence:** 3

**Summary:**

This paper targets the problem of efficient and high-quality reasoning in LLMs. To address the Echo Chamber Effect caused by homogenized bias during inference, it proposes a novel KV cache compression algorithm that utilizes coverage-optimizing strategies to alleviate the Echo Chamber Effect in LLM reasoning. At the query level, it employs Query Diversification (QD), which modifies the queries by removing shared components and amplifying residual differences, enabling each head to focus on a broader and less overlapping set of tokens. At the head level, it applies the Redundancy-Aware Budget Allocation (RABA) algorithm, which reallocates the token budget across heads in proportion to their distinctiveness, granting larger budgets to heads that capture more unique information. Experiments show that the proposed method achieves performance close to the full KV cache in long-context tasks while greatly reducing the required cache budget.

**Strengths:**

1. The paper proposes a novel KV cache compression algorithm based on coverage-optimizing strategies, achieving notable improvements in long-context scenarios.
2. It conducts comprehensive comparisons against multiple baselines and achieves consistent gains across most settings.
3. The work includes thorough ablation studies and efficiency analyses, demonstrating that the approach maintains strong performance while significantly reducing memory cost.

**Weaknesses:**

1. The experiments are conducted primarily on long-context and needle-in-a-haystack tasks. Additional evaluations on broader benchmarks and diverse application domains would help confirm the generalization capability of the proposed method.
2. The explanation of the RABA algorithm for budget reallocation is somewhat complex. Including a step-by-step illustration or concrete example of how token budgets are adjusted across heads would make the method easier to follow.

**Questions:**

1. Would it be possible to evaluate the proposed method on more general benchmarks and across different model scales (e.g., small, medium, and large models) to better demonstrate its generalization and robustness?
2. Could the authors provide a detailed comparison of how the proposed budget allocation differs from prior methods under identical instruction–response pairs, for example by visualizing the distribution of KV cache usage across heads?

---

> ### Author Response · Authors · 2025-11-24
> **Official Comment by Authors**
>
> We thank the reviewer for the constructive feedback and for highlighting the effectiveness of our coverage optimizing strategies and empirical analysis. We address each point below.
>
> **[Q1] Model Scale Generalization**
>
> We evaluated ApertureKV on small (1B), medium (3B), and large (16B) Instruct models to directly address the reviewer’s question regarding generalization and robustness across model scales.
>
> Across all settings, ApertureKV consistently improves performance over AdaKV [1], demonstrating that our coverage optimizing strategies transfer reliably beyond the original 7B/8B experiments and remain effective across different model capacities and architectures.
>
> **Table. Model (LongBench 6 subtasks)**
>
> | **Model** | **KV Size** | Llama-3.2-1B-Instruct (Small) | Llama-3.2-3B-Instruct (Medium) | DeepSeek-v2-Lite-Chat-16B (Large) |
> | --- | --- | --- | --- | --- |
> | AdaKV (Baseline) | 64 | 16.1 | 25.4 | 29.1 |
> | ApertureKV (Ours) | 64 | **17.3** | **26.5** | **29.9** |
> | AdaKV (Baseline) | 128 | 18.8 | 26.4 | 31.9 |
> | ApertureKV (Ours) | 128 | **19.4** | **27.0** | **32.4** |
>
> **[W2 & Q2] RABA Explanation Clarity & Comparison**
>
> To clarify the reviewer’s question, we provide a step-by-step description of RABA below and point readers to **Appendix Section A (Algorithm 4)** for the full pseudocode and **Appendix Section I** for qualitative visualizations comparing budget allocation before and after RABA.
>
> RABA operates in four simple steps:
>
> - Compute per-head token score distributions $\mathbf{s}_{h}$
>
>     RABA first computes diversified attention matrices $\mathbf{A}^{\mathrm{div}}_h$ from diversified queries $\mathbf{Q}^{\mathrm{div}}_h$ and derives the per-head token-score distribution:
>
>     $$
>     \mathbf{A}^{\mathrm{div}}_h
>     = \mathrm{softmax}\left(
>     \frac{
>     \mathbf{Q}^{\mathrm{div}}_h (\mathbf{K}^{\text{prefix}}_h)^\top
>     }{\sqrt{d_c}}
>     \right),
>     $$
>
>     $$ \mathbf s_h = \mathrm{softmax} \Big({1 \over N_{\text{window}}} \sum_{i=1}^{N_{\text{window}}} \mathbf{A}_{h}^{\mathrm{div}}[i,:]\Big).$$
>
> - Compute initial Top-K budgets $B_{h}$
>
>     RABA gathers all token-score distributions and selects the global Top-$B$, then counts how many selected tokens originate from each head:
>
>     $$
>     B_h
>     = \sum_{t \in \mathrm{Top}\text{-}B(\cup_j \mathbf{s}_j)}
>     \mathbf{1}[\ t \in \text{head } h \].
>     $$
>
> - Compute head distinctiveness $w_{h}$
>
>     RABA measures head-level distinctiveness using Jensen–Shannon Divergence (JSD):
>
>     $$
>     D_h = {1 \over H-1 } \sum_{h' \ne h}\mathrm{JSD}{( \mathbf s_h || \mathbf s_{h^\prime} )},
>     $$
>
>     $$
>     w_h = \frac{D_h}{\sum_j D_j}.
>     $$
>
> - RABA reallocates the per-head budgets $B_h$ under a fixed total budget $B$:
>
>     $$
>     \tilde{B}_h
>     = \left\lfloor
>     \frac{w_h B_h}{\sum_j w_j B_j}
>     \cdot B
>     \right\rfloor,
>     $$
>
>     $$
>     \sum_{h=1}^{H} \tilde{B}_h = B.
>     $$
>
>
> This formulation naturally preserves the total budget $B$ while increasing budgets for more distinctive heads and decreasing them for redundant heads.
>
> **Reference**
>
> [1] Feng, Yuan, et al. “AdaKV: Optimizing KV Cache Eviction by Adaptive Budget Allocation for Efficient LLM Inference.” (NeurIPS 2025)

---

### Official Review · Reviewer_rWAv · 2025-10-30

**Soundness:** 3
**Presentation:** 2
**Contribution:** 4
**Rating:** 6
**Confidence:** 4

**Summary:**

This paper introduces ApertureKV, a KV cache compression method that reduces redundancy between tokens retained in the cache. ApertureKV makes diversified query vectors by shifting them away from their centroid, then uses these diversified queries to compute attention scores for selecting tokens for the cache. In addition, ApertureKV saves more tokens for heads that have more distinct token score distributions compared to other heads. With these 2 techniques, ApertureKV achieves state-the-art performance on the LongBench, LooGLE, and Needle-in-a-Haystack benchmarks.

**Strengths:**

1. ApertureKV has state-of-the-art performance on 3 benchmarks with 2 different models, with the best F1 score on every evaluated KV cache size except 2 cases.
2. The paper demonstrates empirically that query redundancy and head redundancy occurs in existing methods, forming a strong basis for its motivation.

**Weaknesses:**

1. The paper claims that baselines produce logically inconsistent answers, but some examples in Figure 7 show that the baseline answers are verbose but correct. The baselines may unnecessarily repeat the answer or the question, but it should not affect task accuracy. If the goal is simply to shorten model outputs, there may be simpler effective methods than ApertureKV. This undercuts the motivation of ApertureKV.
2. While the paper shows that query redundancy and head redundancy occur in baselines, it does not show that they are responsible for the verbose outputs. This causation is assumed by the paper and weakens the motivating logic.

**Questions:**

Questions:
1. In Table 4, why does performance start dropping when the diversification coefficient drops from 0.45 -> 0.50? Does it continue dropping at 0.55? Why was 0.45 tested when all other values were chosen at intervals of 0.1?

Suggestions:
1. Appendix B should include direct empirical evidence that ApertureKV addresses redundancy in the KV cache, by showing correlations between KV values in baseline caches vs. ApertureKV’s cache. ApertureKV’s token score distribution should be shown in Figure 5 to demonstrate that ApertureKV resolves the redundancy issue shown.
2. Using 6.5% as the max KV cache size seems to focus the evaluation only on very small KV cache sizes. The advantage of ApertureKV in Table 1 is highest for the smallest KV sizes, becoming more comparable at KV size 2048 (6.5%). It would be good to add evaluation at bigger KV caches like 4096 (13%), to demonstrate that ApertureKV’s advantage does not disappear at more moderate cache sizes.
3. Can all baselines be included in Table 3? Especially HeadKV, which is the highest performing baseline for the chosen setting. The efficiency comparison would be valuable to show that computing query diversification and redundancy-aware budget allocation does not add substantial overhead. Alternatively, the overhead of QD and RABA could be shown in a separate evaluation that turns them off.
4. The related work on KV Cache Compression should include KVZip[1], a recent KV cache compression method that outperforms previous methods.

Minor comments:
1. Figure 1 does not show much information. It only shows that the baseline contains similar things in its KV cache whereas ApertureKV contains different “things”, but in a way that is too abstract to convey anything deeper. Perhaps it could be replaced with a figure that visualizes the actual KV caches (showing higher correlations within the baseline KV cache and lower correlations in ApertureKV), or the results shown in (C) should just be made bigger.
2. Figure 3 should include a color legend to make clear what red vs green represents.
3. The y-axis label in Figure 6 should say “Pairwise cosine similarity” instead of “Value”, which provides no information.
4. “Echo Chamber Effect” is a cool name but perhaps oversells the issue. “Echo chamber” seems to imply a cycle of compounding error, but the issue at hand is simple redundancy.

[1] Kim et al. “KVzip: Query-Agnostic KV Cache Compression with Context Reconstruction.” NeurIPS 2025.

---

> ### Author Response · Authors · 2025-11-24
> **Official Comment by Authors**
>
> We thank the reviewer for the thorough evaluation and for highlighting the empirical grounding and benchmark performance of ApertureKV. We address each point below.
>
> **[W1] Why verbosity reduces f1: impact on accuracy**
>
> We clarify that even when a baseline output is “verbose but correct” verbosity directly reduces F1 score under token-level evaluation. In free-form QA tasks, F1 score is computed from the predicted token set $P$ and ground-truth token set $G$. The evaluation metrics are:
>
> $
> \begin{equation*}
> \text{precision} = \frac{|P \cap G|}{|P|}, \quad \text{recall} = \frac{|P \cap G|}{|G|},\quad \text{F1 score} = \frac{2 \cdot \text{precision} \cdot \text{recall}}{\text{precision} + \text{recall}}.
> \end{equation*}
> $
>
> Verbose outputs increase the prediction length $|P|$ by repeating or adding irrelevant tokens, while the true overlap $|P \cap G|$ remains unchanged. This necessarily decreases precision:
>
> $
> |P|\uparrow \rightarrow \text{precision}\downarrow \rightarrow \text{F1 score}\downarrow.
> $
>
> Thus, even if an answer appears correct, verbosity reduces F1. More details are provided in Appendix Section H.
>
> **[W2 & S1]  Echo Chamber Effect — Evidence & Mitigation**
>
> We provide additional quantitative analyses that the redundancy induced by top-k KV compression (query redundancy, head redundancy, and low coverage) leads to incorrect, verbose or even repetitive outputs, which lowers F1 score. As briefly discussed in the global response, our ApertureKV mitigates this by reducing redundancy and increasing coverage under the same KV budget (KV size = 64), resulting in up to the improvement of 5.55 in F1 score with Mistral-7B-Instruct. More details are provided in Appendix F.
>
> **Table. Evidence Linking Redundancy to Coverage and Output Quality (LongBench 6 subtasks)**
>
> | **Metric** | **KV size** | **Llama3-8B-Instruct (Baseline)** | **Llama3-8B-Instruct (Ours)** | **Mistral-7B-Instruct (Baseline)** | **Mistral-7B-Instruct (Ours)** |
> | --- | --- | --- | --- | --- | --- |
> | Query Redundancy (↓) | 64 | 0.76 | **0.59** | 0.78 | **0.61** |
> | Head Redundancy (↓) | 64 | 0.35 | **0.29** | 0.34 | **0.31** |
> | Coverage (%) (↑) | 64 | 4.53 | **6.42** | 3.47 | **8.98** |
> | F1 score (↑) | 64 | 26.28 | **28.03** | 22.91 | **28.46** |
> |  |  |  |  |  |  |
>
> **[Q1] Full sweep of the diversification coefficient $\lambda$**
>
> We clarify that the earlier sweep of $\lambda$ was limited by time. After conducting a full sweep (shown below), we find that ApertureKV remains stable across the practical range 0.10–0.90, exhibiting only mild fluctuations. Importantly, all values in this range **consistently outperform the baseline setting ($\lambda=0$)**. The small drop from 0.45 → 0.50 is simply a minor variation rather than the beginning of a downward trend.
>
> **Table. Analysis on Diversification Coefficient (Mistral-7B, LongBench (6 subtasks), KV size = 64)**
>
> | **Method** | $\lambda$ = 0     | 0.05  | 0.10  | 0.15      | 0.20  | 0.25  | 0.30  | 0.35  | 0.40  | 0.45      | 0.50  | 0.55  | 0.60  | 0.65  | 0.70  | 0.75  | 0.80  | 0.85  | 0.90  | 0.95  | 1.00  |
> | ---------- | ----- | ----- | ----- | --------- | ----- | ----- | ----- | ----- | ----- | --------- | ----- | ----- | ----- | ----- | ----- | ----- | ----- | ----- | ----- | ----- | ----- |
> | **+ QD**   | 22.91 | 24.21 | 24.84 | **26.91** | 26.48 | 26.21 | 25.68 | 26.65 | 26.32 | 26.69 | 25.96 | 26.05 | 26.16 | 26.10 | 26.60 | 26.32 | 26.12 | 25.79 | 24.71 | 24.40 | 24.13 |
>
> **[S2] Larger KV Budgets (KV size = 4096, 13%)**
>
> ApertureKV shows **almost no additional gain at 4096 KV size**, because—as already visible at KV Size = 2048 in the main table—**the baseline approaches FullKV performance**. Once the KV budget becomes large enough, redundancy is no longer the bottleneck, so coverage optimizing strategies contribute little further improvement.
>
> **Table. Performance Saturation at Large KV Budgets (LongBench 6 subtasks)**
>
> | **Method** | **KV Size** | **Mistral-7B-Instruct** |
> | --- | --- | --- |
> | FullKV | 31500 (100%) | **32.98** |
> | AdaKV  | 2048 (6.5%) | 32.45 |
> | ApertureKV | 2048 (6.5%) | 32.90 |
> | ApertureKV | 4096 (13%) | 32.92 |
>
> **[S4] Missing Related Work (KVZip)**
>
> Thank you for the pointer. We will include KVZip [2] in the related-work section.
>
> **Reference**
>
> [1] Feng, Yuan, et al. “AdaKV: Optimizing KV Cache Eviction by Adaptive Budget Allocation for Efficient LLM Inference.” (NeurIPS 2025)
>
> [2] Kim, J. H., Kim, J., Kwon, S., Lee, J. W., Yun, S., & Song, H. O. “KVzip: Query-Agnostic KV Cache Compression with Context Reconstruction.” (NeurIPS 2025)

---

> > ### Author Response · Authors · 2025-12-01
> > **Official Comment by Authors - 2**
> >
> > **[S3] Efficiency Comparison with HeadKV and Overhead of QD/RABA**
> >
> > We thank the reviewer for the valuable suggestion. The reviewer requests (1) including **all baselines**, especially **HeadKV**, in the efficiency comparison and (2) explicitly demonstrating that **Query Diversification (QD)** and **Redundancy-Aware Budget Allocation (RABA)** do not introduce overhead.
> >
> > To address this, we extend our latency evaluation to include **HeadKV, AdaKV**, and all **ApertureKV variants (QD, RABA, QD+RABA)** under identical model, prefill, and decode settings. All methods generate the same fixed output lengths (8, 16, 32, and 64 tokens), and we report **per-sample latency (excluding model loading time)** for the prefill, decode, and end-to-end stages.
> >
> > **Key findings:**
> >
> > - **QD-only** exhibits prefill and decode latencies that are **empirically identical** to HeadKV and AdaKV.
> > - **RABA-only** adds a **minor prefill overhead** from per-head allocation, while **decode latency remains unchanged** compared to HeadKV and AdaKV.
> > - **QD+RABA** achieves **end-to-end latency essentially identical to HeadKV and AdaKV**.
> >
> > Taken together, these results show that **QD and RABA introduce negligible overhead** and that ApertureKV maintains **the same practical inference efficiency** as existing baselines while providing substantial accuracy gains.
> >
> > The full latency tables are shown below, and Appendix K provides the detailed experimental setup and additional visualizations.
> >
> > **Table. Llama3-8B-Instruct (KV size = 64).**
> >
> > |Method|Stage|8 tokens|16 tokens|32 tokens|64 tokens|
> > |-|-|-|-|-|-|
> > |HeadKV|Prefill|0.8655|0.8308|0.7714|0.7863|
> > ||Decode|0.2749|0.5600|1.1032|2.1035|
> > ||E2E|1.1404|1.3908|1.8746|2.8898|
> > |AdaKV|Prefill|0.8871|0.8399|0.7818|0.7936|
> > ||Decode|0.2795|0.5773|1.1396|2.1392|
> > ||E2E|1.1666|1.4172|1.9214|2.9328|
> > |ApertureKV (QD)|Prefill|0.8868|0.8505|0.7661|0.7116|
> > ||Decode|0.2833|0.5729|1.1504|2.2221|
> > ||E2E|1.1701|1.4234|1.9165|2.9337|
> > |ApertureKV (RABA)|Prefill|0.8933|0.8522|0.8185|0.7647|
> > ||Decode| 0.2857|0.5755|1.1032|2.1834|
> > ||E2E|1.1790|1.4277|1.9217|2.9481|
> > |ApertureKV (QD+RABA)|Prefill|0.8967|0.8646|0.7785|0.6659|
> > ||Decode|0.2712|0.5533|1.1481| 2.2729|
> > ||E2E|1.1679|1.4179|1.9266|2.9388|
> >
> > **Table. Llama3-8B-Instruct (KV size = 128).**
> >
> > |Method| Stage | 8 tokens | 16 tokens | 32 tokens | 64 tokens |
> > |-|-|-|-|-|-|
> > |HeadKV|Prefill|0.8889|0.8567|0.8089|0.7183|
> > ||Decode|0.2697|0.5346|1.0579|2.1124|
> > ||E2E|1.1586|1.3913|1.8668|2.8307|
> > |AdaKV|Prefill|0.9076|0.8814|0.8195|0.7952|
> > ||Decode|0.2672|0.5377|1.0663|2.0985|
> > ||E2E |1.1748|1.4191|1.8858|2.8937|
> > |ApertureKV (QD)|Prefill|0.9167|0.8868|0.8417|0.7932|
> > ||Decode|0.2667|0.5291|1.0485|2.0778|
> > ||E2E|1.1834|1.4159|1.8902|2.8710|
> > |ApertureKV (RABA)| Prefill |0.9281|0.9089|0.9119|0.8822|
> > ||Decode|0.2566|0.5123|0.9999| 2.0174|
> > ||E2E|1.1847|1.4212|1.9118| 2.8996|
> > |ApertureKV (QD+RABA)|Prefill|0.9170|0.8901|1.0247|0.8532|
> > ||Decode|0.2497|0.5133|1.0593|2.0374|
> > ||E2E|1.1667|1.4034|2.0840|2.8906|
> >
> > **Table. Mistral-7B-Instruct (KV size = 64).**
> >
> > |Method|Stage|8 tokens|16 tokens|32 tokens|64 tokens|
> > |-|-|-|-|-|-|
> > |HeadKV|Prefill|1.9325|1.8872|1.8172|1.7040|
> > ||Decode|0.3481|0.6840|1.3512|2.6946|
> > ||E2E|2.2806|2.5712|3.1684|4.3986|
> > |AdaKV|Prefill|1.9603|1.9107|1.8378|1.6139|
> > ||Decode|0.3376|0.6747|1.3412|2.8118|
> > ||E2E|2.2979|2.5854|3.1790|4.4257|
> > |ApertureKV (QD)|Prefill|1.9563|1.8996|1.8762|1.5391|
> > ||Decode|0.3463|0.6948|1.3582|2.8812|
> > ||E2E|2.3026|2.5944|3.2344|4.4203|
> > |ApertureKV (RABA)|Prefill|1.9652|1.9027|1.8660|1.6068|
> > ||Decode|0.3468|0.6961|1.3807|2.8211|
> > ||E2E| 2.3120|2.5988|3.2467|4.4279|
> > |ApertureKV (QD+RABA)|Prefill|2.0079|1.9608|1.8983|1.5927|
> > ||Decode|0.3477|0.6931|1.3612|2.8638|
> > ||E2E| 2.3556|2.6539|3.2595|4.4565|
> >
> > **Table. Mistral-7B-Instruct (KV size = 128).**
> >
> > |Method|Stage|8 tokens|16 tokens|32 tokens|64 tokens|
> > |-|-|-|-|-|-|
> > |HeadKV|Prefill|1.9947|1.9611|1.8356|1.6328|
> > ||Decode|0.3342|0.6657|1.3750|2.7616|
> > ||E2E|2.3289|2.6268|3.2106|4.3944|
> > |AdaKV|Prefill|2.0037|1.9722|1.9049|1.7089|
> > ||Decode|0.3385|0.6667|1.3186|2.6861|
> > ||E2E|2.3422|2.6389|3.2235|4.3950|
> > |ApertureKV (QD)|Prefill|2.0219|1.9830|1.8785|1.7409|
> > ||Decode|0.3220|0.6708|1.3488|2.6740|
> > ||E2E|2.3439|2.6538|3.2273|4.4149|
> > |ApertureKV (RABA)|Prefill|2.0267|1.9828|1.9060|1.7843|
> > ||Decode|0.3275|0.6797|1.3378| 2.6544|
> > ||E2E|2.3542|2.6625|3.2438|4.4387|
> > |ApertureKV (QD+RABA)| Prefill|2.0257|1.9879|1.9661|1.7966|
> > ||Decode|0.3325|0.6628|1.2807|2.6467|
> > ||E2E|2.3582|2.6507|3.2468|4.4433|

---

### Official Review · Reviewer_YpMQ · 2025-10-31

**Soundness:** 3
**Presentation:** 2
**Contribution:** 2
**Rating:** 4
**Confidence:** 4

**Summary:**

The paper identifies a new pathology in top-K KV-cache compression, the “Echo-Chamber Effect”: successive attention heads and queries repeatedly select near-identical tokens, so the compressed cache suffers low coverage and long-range reasoning degrades.
To counter this the authors propose ApertureKV, a prefill-only method with two synergistic modules:
(1) Query Diversification (QD) that subtracts a shared centroid from windowed queries, and
(2) Redundancy-Aware Budget Allocation (RABA) that re-allocates the per-head token budget proportionally to the Jensen-Shannon divergence between head score distributions.
Extensive experiments on LongBench, LooGLE and Needle-in-a-Haystack (Llama-3-8B, Mistral-7B) show that ApertureKV retains ≥ 92 % of full-cache F1 with only 0.2 % of the KV memory, outperforming SnapKV, PyramidKV, HeadKV and Ada-KV while keeping the same runtime and VRAM. Ablation studies and qualitative examples confirm that coverage optimization, not just higher cache size, drives the gains.

**Strengths:**

1. It ntroduces a new problem, echo-chamber effect, and two complementary, easy-to-implement fixes.
2. It clearly separates query-level vs head-level redundancy and shows both must be addressed.
3. Evaluation on 11 long-context tasks, two models, four cache sizes, plus ablations and significance tests.

**Weaknesses:**

1.	The paper lacks an analysis of the echo-chamber effect and does not provide solid evidence that the proposed method can effectively address this issue.
2.	Experiments are restricted to Llama-3-8B-Instruct and Mistral-7B-Instruct, two models that supports a finite context length of only 7,950 and 31,500 separately. I doubt whether conducting experiments on long context benchmarks with the two models can yield accurate results.

**Questions:**

1.	How does ApertureKV behave with grouped-query attention (GQA) or multi-query attention (MQA) where head redundancy is structurally reduced?
2.	According to the authors, QD can eliminate the Echo-Chamber Effect on full KV. Do other existing compression methods (e.g., adaKV) also have this effect? If so, is further analysis needed on the causes of this phenomenon?
3.	Could the authors list the text lengths of the tasks in the benchmarks and compare them with the model's context window length? If the window length is insufficient, should extra experiments be conducted on models with longer context support?

---

> ### Author Response · Authors · 2025-11-24
> **Official Comment by Authors**
>
> We thank the reviewer for thorough assessment, especially focusing on the formulation of the Echo Chamber Effect and the roles of QD and RABA. We address each question below.
>
> **[Q1] Generalization to Attention Architecture (GQA/MQA)**
>
> Great question! While our formulations of QD and RABA in Eq. (8-17) of the main paper are designed for the standard Multi-Head Attention (MHA), they are directly applicable to GQA, MQA, and MLA without any modification. The discussions and equations in **Appendix G confirm that** our methods are independent from how key and value are shared or projected.
>
> Moreover, additional experiments across *“four”* different language models show that our ApertureKV (QD+RABA) consistently improves the performance of GQA and MLA even with various KV sizes.
>
> **Table. Performance on GQA and MLA Architectures (LongBench, 6 subtasks) (provided in Appendix Section J).**
>
> |**Method**|**KV size**|**Llama-3-8B-Instruct (GQA)**|**Mistral-7B-Instruct (GQA)**|**DeepSeekv2-Lite-16B (MLA)**| **DeepSeekv2-Lite-Chat-16B (MLA)**|
> |-|-|-|-|-|-|
> |AdaKV (Baseline)|64|26.2|22.9|16.4|29.1|
> |ApertureKV (Ours)|64|**28.0**|**28.5**|**17.2**|**29.9**|
> |AdaKV (Baseline)|128|28.6 |25.8|17.0|31.9|
> |ApertureKV (Ours)|128|**30.0**|**29.8**|**17.5**|**32.4**|
>
> Although we theoretically show that our method readily extendable to MQA, at the moment, no reasonable open-source MQA-based LLM is publicly available. If it becomes available, we will include the empirical results in the final version.
>
> **[W1 & Q2] Echo Chamber Effect - Evidence & Mitigation**
>
> We first clarify that our analysis of the Echo Chamber Effect concerns redundancy introduced by top-k KV compression, not redundancy in the full KV cache. In existing compression baselines such as AdaKV [1], the compressed KV cache repeatedly selects similar tokens across queries and heads, resulting in high redundancy and low effective coverage. Importantly, ApertureKV does not modify the full KV: QD adjusts only the *queries used during compression*, and RABA redistributes the *compression budget across heads.*
>
> To examine whether existing approaches (e.g., AdaKV) mitigate the Echo Chamber Effect, we measured redundancy directly on their compressed KV caches and compared them with ApertureKV under the same KV budget (KV size = 64). We observed that baseline still exhibits (1) high query redundancy, (2) high head redundancy, and (3) low coverage, indicating that existing compression methods do not eliminate this phenomenon. This confirms that further analysis is necessary and directly addresses the reviewer’s question.
>
> **Table. Quantitative Evidence of the Echo Chamber Effect (LongBench, 6 subtasks)**
>
> | **Metric** | **KV size** | **Llama3-8B-Instruct (Baseline)** | **Llama3-8B-Instruct (Ours)** | **Mistral-7B-Instruct (Baseline)** | **Mistral-7B-Instruct (Ours)** |
> | --- | --- | --- | --- | --- | --- |
> | Query Redundancy (↓) | 64 | 0.76 | **0.59** | 0.78 | **0.61** |
> | Head Redundancy (↓) | 64 | 0.35 | **0.29** | 0.34 | **0.31** |
> | Coverage (%) (↑) | 64 | 4.53 | **6.42** | 3.47 | **8.98** |
> | F1 score (↑) | 64 | 26.28 | **28.03** | 22.91 | **28.46** |
> |  |  |  |  |  |  |
>
> These results show that ApertureKV significantly reduces query-level and head-level redundancy while increasing coverage, directly mitigating the Echo Chamber Effect under the same KV budget (KV size = 64). Detailed formulas and explanations for these metrics are provided in Appendix F.
>
> **[W2 & Q3] Benchmark sequence length vs. model context limitation**
>
> In the main paper, experiments are conducted with 8k/32k-context models, which inevitably lead to context truncation on several subtasks. To validate the effectiveness of our method under no context-truncation scenarios, we provide additional experiments using 128k-context models that can handle all subtasks of LongBench and LooGLE  with longer sequence lengths (10k ~ 40k). Our experiments show that ApertureKV consistently outperforms AdaKV [1], demonstrating that our method remains effective even when the entire sequence is accessible. The maximum sequence lengths for all subtasks are explicitly summarized in Appendix J (Table 10)
>
> **Table. LongBench (6 subtasks): F1 score on 128k-Context Models.**
>
> | **Context Length** | **Method** | **KV Size** | **Llama-3.2-1B-Instruct** | **Llama-3.2-3B-Instruct** |
> | --- | --- | --- | --- | --- |
> | 128k | AdaKV (Baseline) | 64 | 16.1 | 25.4 |
> | 128k | ApertureKV (Ours) | 64 | **17.3** | **26.5** |
> | 128k | AdaKV (Baseline) | 128 | 18.8 | 26.4 |
> | 128k | ApertureKV (Ours) | 128 | **19.4** | **27.0** |
>
>  **Table. LooGLE (4 subtasks): F1 score on 128k-Context Models.**
>
> | **Context Length** | **Method** | **KV Size** | **Llama-3.2-1B-Instruct** | **Llama-3.2-3B-Instruct** |
> | --- | --- | --- | --- | --- |
> | 128k | AdaKV (Baseline) | 64 | 7.1 | 8.1 |
> | 128k | ApertureKV (Ours) | 64 | **7.3** | **9.2** |
> | 128k | AdaKV (Baseline) | 128 | 7.8 | 9.3 |
> | 128k | ApertureKV (Ours) | 128 | **8.2** | **10.1** |

---

### Official Review · Reviewer_ZxzQ · 2025-11-01

**Soundness:** 3
**Presentation:** 3
**Contribution:** 2
**Rating:** 4
**Confidence:** 4

**Summary:**

The paper identifies a limitation in existing KV cache compression methods, which typically assess cache importance using a unary importance score. This approach results in a homogeneous compressed KV cache and consequently reduces diversity. To address this issue, the authors propose ApertureKV, a method that preserves KV cache entries using diversified queries and employs a redundancy-aware budget allocation strategy to improve compression effectiveness.

**Strengths:**

The paper presents a clear motivation and a well-defined solution, supported by clear writing and presentation. The proposed method outperforms existing baselines across a diverse set of benchmarks.

**Weaknesses:**

The paper lacks a discussion on the compression cost, raising concerns about the practicality of the proposed method in deployment scenarios. Since it utilizes much more queries than baselines such as SnapKV, the end-to-end inference time may increase substantially, potentially making the method impractical for real-world use. It would strengthen the paper to include an analysis of the end-to-end inference speed, varying the prefill and decode sizes, to better assess the actual runtime efficiency.

**Questions:**

See the above weakness section.

**Details Of Ethics Concerns:**

I do not have ethics concern regarding this paper.

---

> ### Author Response · Authors · 2025-12-01
> **Official Comment by Authors**
>
> **[W1] End-to-End Latency & Number of Window Queries**
>
> We thank the reviewer for raising the concern about **compression cost** and the potential impact on **end-to-end inference time**. The reviewer’s question centers on whether ApertureKV uses more queries than prior methods and whether this could harm **runtime efficiency** in deployment.
>
> **We clarify that ApertureKV does *not* use more queries than SnapKV, PyramidKV, HeadKV, or AdaKV.**
>
> All methods—including ours—use the same fixed set of **8 window queries**, which are the *only* queries used during the prefill stage to **score past tokens and select which KV entries to retain** before decoding begins. ApertureKV applies the exact same mechanism and does not introduce any additional queries, ensuring **no query-related overhead**.
>
> To directly address the reviewer’s request, we evaluate the **end-to-end inference speed** for five methods (HeadKV, AdaKV, ApertureKV-QD, ApertureKV-RABA, ApertureKV-QD+RABA) under identical model, prefill, and decode settings. We report **per-sample latency (excluding model loading time)** for the prefill, decode, and overall E2E stages while varying both **prefill settings (KV sizes = 64 and 128)** and **decode lengths (fixed generation lengths = 8, 16, 32, 64 tokens)**.
>
> Across all configurations, we find that
>
> - **decode latency (the dominant component of E2E time) is nearly identical** across all methods,
> - **QD adds no measurable overhead**, and
> - **RABA introduces only a small constant prefill cost**.
>
> These results demonstrate that ApertureKV achieves **end-to-end inference time comparable to existing methods**, confirming its practicality for real-world deployment.
>
> The full latency tables are shown below, and Appendix K provides the detailed experimental setup and additional visualizations.
>
> **Table. Llama3-8B-Instruct (KV size = 64).**
>
> |Method|Stage|8 tokens|16 tokens|32 tokens|64 tokens|
> |-|-|-|-|-|-|
> |HeadKV|Prefill|0.8655|0.8308|0.7714|0.7863|
> ||Decode|0.2749|0.5600|1.1032|2.1035|
> ||E2E|1.1404|1.3908|1.8746|2.8898|
> |AdaKV|Prefill|0.8871|0.8399|0.7818|0.7936|
> ||Decode|0.2795|0.5773|1.1396|2.1392|
> ||E2E|1.1666|1.4172|1.9214|2.9328|
> |ApertureKV (QD)|Prefill|0.8868|0.8505|0.7661|0.7116|
> ||Decode|0.2833|0.5729|1.1504|2.2221|
> ||E2E|1.1701|1.4234|1.9165|2.9337|
> |ApertureKV (RABA)|Prefill|0.8933|0.8522|0.8185|0.7647|
> ||Decode| 0.2857|0.5755|1.1032|2.1834|
> ||E2E|1.1790|1.4277|1.9217|2.9481|
> |ApertureKV (QD+RABA)|Prefill|0.8967|0.8646|0.7785|0.6659|
> ||Decode|0.2712|0.5533|1.1481| 2.2729|
> ||E2E|1.1679|1.4179|1.9266|2.9388|
>
> **Table. Llama3-8B-Instruct (KV size = 128).**
>
> |Method| Stage | 8 tokens | 16 tokens | 32 tokens | 64 tokens |
> |-|-|-|-|-|-|
> |HeadKV|Prefill|0.8889|0.8567|0.8089|0.7183|
> ||Decode|0.2697|0.5346|1.0579|2.1124|
> ||E2E|1.1586|1.3913|1.8668|2.8307|
> |AdaKV|Prefill|0.9076|0.8814|0.8195|0.7952|
> ||Decode|0.2672|0.5377|1.0663|2.0985|
> ||E2E |1.1748|1.4191|1.8858|2.8937|
> |ApertureKV (QD)|Prefill|0.9167|0.8868|0.8417|0.7932|
> ||Decode|0.2667|0.5291|1.0485|2.0778|
> ||E2E|1.1834|1.4159|1.8902|2.8710|
> |ApertureKV (RABA)| Prefill |0.9281|0.9089|0.9119|0.8822|
> ||Decode|0.2566|0.5123|0.9999| 2.0174|
> ||E2E|1.1847|1.4212|1.9118| 2.8996|
> |ApertureKV (QD+RABA)|Prefill|0.9170|0.8901|1.0247|0.8532|
> ||Decode|0.2497|0.5133|1.0593|2.0374|
> ||E2E|1.1667|1.4034|2.0840|2.8906|
>
> **Table. Mistral-7B-Instruct (KV size = 64).**
>
> |Method|Stage|8 tokens|16 tokens|32 tokens|64 tokens|
> |-|-|-|-|-|-|
> |HeadKV|Prefill|1.9325|1.8872|1.8172|1.7040|
> ||Decode|0.3481|0.6840|1.3512|2.6946|
> ||E2E|2.2806|2.5712|3.1684|4.3986|
> |AdaKV|Prefill|1.9603|1.9107|1.8378|1.6139|
> ||Decode|0.3376|0.6747|1.3412|2.8118|
> ||E2E|2.2979|2.5854|3.1790|4.4257|
> |ApertureKV (QD)|Prefill|1.9563|1.8996|1.8762|1.5391|
> ||Decode|0.3463|0.6948|1.3582|2.8812|
> ||E2E|2.3026|2.5944|3.2344|4.4203|
> |ApertureKV (RABA)|Prefill|1.9652|1.9027|1.8660|1.6068|
> ||Decode|0.3468|0.6961|1.3807|2.8211|
> ||E2E| 2.3120|2.5988|3.2467|4.4279|
> |ApertureKV (QD+RABA)|Prefill|2.0079|1.9608|1.8983|1.5927|
> ||Decode|0.3477|0.6931|1.3612|2.8638|
> ||E2E| 2.3556|2.6539|3.2595|4.4565|
>
> **Table. Mistral-7B-Instruct (KV size = 128).**
>
> |Method|Stage|8 tokens|16 tokens|32 tokens|64 tokens|
> |-|-|-|-|-|-|
> |HeadKV|Prefill|1.9947|1.9611|1.8356|1.6328|
> ||Decode|0.3342|0.6657|1.3750|2.7616|
> ||E2E|2.3289|2.6268|3.2106|4.3944|
> |AdaKV|Prefill|2.0037|1.9722|1.9049|1.7089|
> ||Decode|0.3385|0.6667|1.3186|2.6861|
> ||E2E|2.3422|2.6389|3.2235|4.3950|
> |ApertureKV (QD)|Prefill|2.0219|1.9830|1.8785|1.7409|
> ||Decode|0.3220|0.6708|1.3488|2.6740|
> ||E2E|2.3439|2.6538|3.2273|4.4149|
> |ApertureKV (RABA)|Prefill|2.0267|1.9828|1.9060|1.7843|
> ||Decode|0.3275|0.6797|1.3378| 2.6544|
> ||E2E|2.3542|2.6625|3.2438|4.4387|
> |ApertureKV (QD+RABA)| Prefill|2.0257|1.9879|1.9661|1.7966|
> ||Decode|0.3325|0.6628|1.2807|2.6467|
> ||E2E|2.3582|2.6507|3.2468|4.4433|

---

### Author Response · Authors · 2025-11-24
**Global Response to Reviewers**

We sincerely thank all reviewers for their constructive and detailed feedback.

Our paper aims to **mitigate the Echo Chamber Effect in KV cache compression**—a phenomenon in which top-k selection repeatedly retains **similar (or redundant) tokens** across queries and heads, leading to **limited coverage** under tight KV cache budgets.

To address this limitation, we introduce **ApertureKV**, a KV cache compression approach composed of two **coverage optimization strategies.** The following numbers report improvements on the Mistral-7B-Instruct model (KV size = 64) evaluated over six LongBench subtasks:

- **Query Diversification (QD)**: *by diversifying the queries used during KV cache compression*, **QD** reduces query-level redundancy by 21.8%, increases coverage by 8.7%, and improves F1 score by 16.5%.
- **Redundancy-Aware Budget Allocation (RABA)**: *by reallocating token budgets toward more distinctive heads*, **RABA** reduces head-level redundancy by 8.8%, improves coverage by 13.5%, and improves F1 score by 12.5%.
- **ApertureKV (QD + RABA):** jointly expands coverage from 3.47 **→ 8.98 (158.79%)** and raises F1 score from **22.91 → 28.46 (+24.2%).**

|Method | Query Redundancy ($\downarrow$)  | Head Redundancy ($\downarrow$) |Coverage ($\uparrow$) |F1 score ($\uparrow$) |
| --- | --- | --- | --- | --- |
| Baseline | 0.78 | 0.34 | 3.47 | 22.91  |
| + QD | 0.61 (21.8% gain) | - | 3.77 (8.7% gain) | 26.69 (16.5% gain) |
| + RABA | - | 0.31 (8.8% gain) | 3.94 (13.5% gain) | 25.78 (12.5% gain) |
| ApertureKV (QD + RABA) | - | - | **8.98 (158.8% gain)** | **28.46 (24.2% gain)** |

In particular, **ApertureKV** uses **only 0.2% of the full KV cache** while retaining **92.6% of the original performance**, highlighting its robustness under severe memory constraints.

We briefly address questions raised by multiple reviewers below:

- **Echo Chamber Effect - Evidence & Mitigation [YpMQ, rWAv]**
    - We ***introduce simple metrics for query redundancy, head redundancy, and coverage***, and measure them directly on compressed KV caches. Across LongBench subtasks, ApertureKV **reduces both types of redundancy and significantly improves coverage** under the same KV cache budget, which in turn correlates with higher F1 scores, as detailed in **Appendix F (Tables 6–8).**
- **Generalization to other attention architectures [YpMQ]**
    - We ***analytically verified*** that our method QD/RABA can be applicable to “three” attention modules: **GQA/MQA/MLA**. Moreover, our additional experiments ***empirically validate*** that the proposed method consistently improves GQA & MLA across “four” models (**Llama-3-8B, Mistral-7B, DeepSeekv2-Lite-16B, DeepSeekv2-Lite-Chat-16B**), **as detailed in Appendix J (Table 9).**
- **Robustness under extended context lengths [YpMQ]**
    - We list full sequence lengths for all subtasks and run LongBench and LooGLE experiments on 128k-context Llama-3.2-1B/3B, confirming that ApertureKV **consistently improves performance** even when no truncation occurs, **as detailed in Appendix J (Table 10-11).**
- **Scalability across model sizes [ucRM]**
    - We evaluate ApertureKV on small (1B), medium (3B), and large (16B) models and observe robust improvements across all model scales, **as detailed in Appendix J (Table 12).**

- **End-to-End Latency & Deployment Practicality [ZxzQ, rWAv]**
  - We conduct a comprehensive prefill, decode, and end-to-end (E2E) latency analysis across two models, two KV sizes, and four generation lengths. **ApertureKV achieves decode and overall E2E latency *on par with* the strongest existing baselines (HeadKV/AdaKV)**, demonstrating that the proposed components do not hinder practical inference efficiency, **as detailed in Appendix K (Figure 10).**

Below, we address the Weaknesses (W), Questions (Q), and Suggestions (S) raised by each reviewer.

---

> ### Author Response · Authors · 2025-12-03
> **Post-Rebuttal Summary: Key Clarifications and Improvements**
>
> We sincerely appreciate the careful attention given to this submission. For convenience, we summarize below the key clarifications and improvements made during the rebuttal, directly responding to the reviewers’ major concerns.
>
> **Addressing Reviewer ZxzQ’s concerns (Rating: 4, Confidence: 4):**
>
> ZxzQ mainly questioned the practical deployability of the method, focusing on compression cost and potential latency overhead. We clarified that ApertureKV uses the same 8 window queries as prior methods and added full prefill/decode/E2E latency experiments showing that QD and RABA introduce no measurable runtime overhead.
>
> - **[W1] Compression cost & E2E latency.** We clarified that ApertureKV uses the same fixed 8 window queries as HeadKV/AdaKV and added full latency experiments (2 models, 2 KV sizes, 4 decode lengths) showing QD/RABA match their prefill, decode, and E2E latency.
>
> **Addressing Reviewer YpMQ’s concerns (Rating: 4, Confidence: 4):**
>
> YpMQ mainly focused on methodological validity and architectural generality (architecture applicability, evidence for the Echo Chamber Effect, long-context robustness), and we addressed these by adding analytical justification, GQA/MLA experiments, redundancy/coverage measurements, and 128k-context evaluations confirming consistent gains without truncation.
>
> - **[Q1] Generalization to GQA/MQA/MLA.** We analytically showed that QD/RABA are architecture-agnostic and added experiments on GQA/MLA models (Llama-3-8B-Instruct, Mistral-7B-Instruct, DeepSeekv2-Lite-16B, DeepSeekv2-Lite-Chat-16B), consistently improving F1 score at KV sizes 64 and 128.
> - **[W1 & Q2] Echo Chamber Effect — evidence & mitigation.** We defined metrics for query redundancy, head redundancy, and coverage on compressed KV caches and showed that ApertureKV reduces redundancy and increases coverage relative to AdaKV, improving F1 under the same KV budget.
> - **[W2 & Q3] Context-window limitations.** We added 128k-context experiments on Llama-3.2-1B/3B and listed sequence lengths for all tasks, showing ApertureKV still outperforms AdaKV even without any truncation.
>
> **Addressing Reviewer rWAv’s concerns (Rating: 6, Confidence: 4):**
>
> rWAv mainly questioned whether redundancy is the primary factor behind degraded performance and whether our motivation is empirically justified. We directly addressed these concerns through explicit F1-metric analysis, quantitative redundancy–coverage evidence, and comprehensive efficiency comparisons against HeadKV and AdaKV.
>
> - **[W1] Verbosity vs F1 score.** We explicitly wrote out the token-level F1 formula and explained that verbose-but-correct answers increase prediction length without increasing overlap, thereby lowering precision and F1 score.
> - **[W2 & S1] Redundancy → coverage → output quality.** We added quantitative evidence that higher query/head redundancy and lower coverage correlate with worse F1, and that ApertureKV reduces redundancy, improves coverage, and raises F1 under the same KV budget.
> - **[Q1] Full sweep of the diversification coefficient.** We conducted a full sweep of the diversification coefficient, showing stable performance over a wide range (0.10–0.90) and consistent gains over the baseline.
> - **[S2] Larger KV budgets.** We reported results at KV size 4096 and showed that performance saturates near FullKV, explaining why ApertureKV’s benefit naturally diminishes when redundancy is no longer the bottleneck.
> - **[S3] Efficiency comparison and overhead analysis.** We extended the latency study to include HeadKV and all ApertureKV variants, confirming that QD incurs no measurable latency and RABA adds only minor prefill overhead with essentially identical E2E time.
> - **[S4] Missing related work (KVZip).** We acknowledged KVZip and committed to including it in the revised related-work section.
>
> **Addressing Reviewer ucRM’s concerns (Rating: 6, Confidence: 3):**
>
> ucRM mainly focused on generalization and algorithmic clarity (scaling behavior, RABA transparency), and we addressed these by adding 1B→3B→16B experiments demonstrating scale robustness and providing a step-by-step RABA breakdown with pseudocode and qualitative visualizations.
>
> - **[Q1] Model-scale generalization.** We added LongBench experiments on 1B, 3B, and 16B models and showed that ApertureKV consistently improves F1 score over AdaKV across all scales and KV sizes.
> - **[W2 & Q2] RABA explanation clarity & comparison.** We provided a step-by-step description of RABA (token score distributions → initial top-K budgets → JSD distinctiveness → budget reallocation) with qualitative budget-allocation visualizations.

---

### Note · Program_Chairs · 2026-01-17
**Submission Desk Rejected by Program Chairs**

The following references in this submission do not refer to real documents and/or have major errors in bibliographic information:

 Yuchen Jiang et al. Mistral: Efficient open-weight llms. Mistral.ai, 2023.